# Bacterial death and TRADD-N domains help define novel apoptosis and immunity mechanisms shared by prokaryotes and metazoans

**Gurmeet Kaur[†], Lakshminarayan M Iyer[†], A Maxwell Burroughs, L Aravind***

Computational Biology Branch, National Center for Biotechnology Information, National Library of Medicine, National Institutes of Health, Bethesda, United States

**\*For correspondence:**
aravind@mail.nih.gov

[†]These authors contributed equally to this work

**Competing interest:** The authors declare that no competing interests exist.

**Abstract** Several homologous domains are shared by eukaryotic immunity and programmed cell-death systems and poorly understood bacterial proteins. Recent studies show these to be components of a network of highly regulated systems connecting apoptotic processes to counter-invader immunity, in prokaryotes with a multicellular habit. However, the provenance of key adaptor domains, namely those of the Death-like and TRADD-N superfamilies, a quintessential feature of metazoan apoptotic systems, remained murky. Here, we use sensitive sequence analysis and comparative genomics methods to identify unambiguous bacterial homologs of the Death-like and TRADD-N superfamilies. We show the former to have arisen as part of a radiation of effector-associated α-helical adaptor domains that likely mediate homotypic interactions bringing together diverse effector and signaling domains in predicted bacterial apoptosis- and counter-invader systems. Similarly, we show that the TRADD-N domain defines a key, widespread signaling bridge that links effector deployment to invader-sensing in multicellular bacterial and metazoan counter-invader systems. TRADD-N domains are expanded in aggregating marine invertebrates and point to distinctive diversifying immune strategies probably directed both at RNA and retroviruses and cellular pathogens that might infect such communities. These TRADD-N and Death-like domains helped identify several new bacterial and metazoan counter-invader systems featuring underappreciated, common functional principles: the use of intracellular invader-sensing lectin-like (NPCBM and FGS), transcription elongation GreA/B-C, glycosyltransferase-4 family, inactive NTPase (serving as nucleic acid receptors), and invader-sensing GTPase switch domains. Finally, these findings point to the possibility of multicellular bacteria-stem metazoan symbiosis in the emergence of the immune/apoptotic systems of the latter.

## Introduction

Evolutionary evidence favors several independent origins for multicellularity in diverse eukaryotic and prokaryotic lineages (*Grosberg and Strathmann, 2007*; *Lyons and Kolter, 2015*; *Kysela et al., 2016*; *Dunin-Horkawicz et al., 2014*). However, the emergence of this life history characteristic is accompanied by specific similarities across phylogenetically distant branches of the tree of life (*Grosberg and Strathmann, 2007*; *Lyons and Kolter, 2015*; *Rokas, 2008*). One such is programmed cell death, cell suicide, or apoptosis of individual cells within the multicellular assemblage (*Ameisen, 2002*; *Vaux, 1993*). Indeed, apoptosis has been observed and studied across multicellular eukaryotes, such as metazoans, fungi, amoebozoan slime molds (e.g., *Dictyostelium*), and plants as well as certain multicellular prokaryotes such as cyanobacteria and actinobacteria (*Ameisen, 2002*; *Yuan and Kroemer, 2010*; *Elmore, 2007*; *Fuchs and Steller, 2011*; *Greenberg, 1996*; *Bidle and Falkowski, 2004*;

*Jiménez et al., 2009*; *Zheng et al., 2013*; *Filippova and Vinogradova, 2017*). It is shown to occur in several biological contexts: (1) in routine development, where it serves as a mechanism for 'sculpting' the body forms of organisms (e.g., inter-digital cell death in tetrapods) (*Chautan et al., 1999*); (2) in responses to stress and environmental insults, where it might help eliminate cells with DNA damage or other defects, like misfolded proteins, that might prove deleterious to the organism (*Adams, 2003*); (3) as part of the immune response to clear moribund cells and limit infections of viruses and other intracellular parasites by eliminating infected cells (*Imre, 2020*; *Hayakawa et al., 2006*); and (4) self-non-self recognition, where it serves to restrict fusion or mergers of non-kin individuals, for instance, in fungal heterokaryon incompatibility or allo-incompatibility in colonial animals like ascidians, bryozoans, cnidarians, and sponges (*Daskalov et al., 2017*; *Glass and Dementhon, 2006*; *Buss, 1990*).

At first sight, the evolution of a suicidal response appears paradoxical as it effectively nullifies the fitness of the cell. However, such a response can be selected for due to the principle of inclusive fitness. Here, the cell undergoing suicide, despite dying, accrues fitness via the benefits its death confers on identical or closely related kin cells in a multicellular assembly (*Bourke, 2014*; *Hamilton, 1964*). This is amply borne out by the biological contexts in which apoptosis occurs, namely those in which the death of individual cells is for the greater good of the multicellular assemblage of kin cells (*Michod and Roze, 2001*). In these contexts, apoptosis is initiated by specific interactions involving the delivery of effectors or sensing of specific stimuli that are best understood in multicellular eukaryotes. These processes include: (1) direct delivery of initiating effectors from outside to cells targeted for death, such as the lysosomal granules bearing perforin and granzymes by CD8+ cytotoxic T-cells in jawed vertebrates (*Podack, 1995*); (2) extrinsic cell-death-initiating signals communicated via the cell-surface receptors (e.g., receptors with Death domains in metazoans; *Figure 1A*; *Schulze-Osthoff et al., 1998*); (3) intrinsic stimuli sensed by intracellular receptors such as pathogen molecules detected by LRR repeats in plants or the RIG-I viral RNA-sensing helicase module in animals (*Chattopadhyay and Sen, 2017*; *Ting et al., 2008*); and (4) stimuli in the form of molecules sensed at the interface between the mitochondria and the cytoplasm that signal deleterious metabolic (especially oxidative) stress (*Figure 1B*; *Altman and Rathmell, 2012*). Such triggers result in the deployment of effectors that enzymatically target cellular molecules leading to suicide, such as caspases that cleave proteins, DNases that cleave genomic DNA, TIR domains that cleave NAD$^+$, or ADP-ribosyltransferases (ARTs) that modify DNA or proteins (*Adams, 2003*). Alternatively, effector action occurs through membrane perforation by transmembrane proteins (Bcl-2) or non-covalent protein-templated (prion-like) assembly of filamentous polymeric complexes (e.g., of CARD, Pyrin, and TIR domains) within cells (*Figure 1C and D*; *Delbridge et al., 2016*; *Li et al., 2012*; *Gentle et al., 2017*; *Morehouse et al., 2020*).

Despite apoptosis contributing to the greater good of a cooperative assemblage of cells, the unleashing of lethal effectors can be potentially deleterious to an organism if deployed under the wrong circumstances. Thus, the evolution of apoptosis has gone hand-in-hand with a striking array of negative regulators and switches that limits the option of apoptosis to specific situations. In their simplest form, these regulatory interactions involve a pair of paralogous pro-apoptotic and anti-apoptotic proteins, for example, members of the Bcl-2 family of transmembrane (TM) proteins (*Cory et al., 2003*; *Hawkins and Vaux, 1994*). Increasingly complex forms of regulation are constituted by well-coordinated cascades of enzyme activity (e.g., proteolytic cascades mediated by the caspases and ZU5 domains) or energy-requiring, threshold-dependent switches mediated by different families of NTPases (*Allen et al., 1998*; *Janssens and Tinel, 2012*; *D'Osualdo et al., 2011*; *Zhang et al., 2012*). Regulation is also achieved by a series of shifting non-covalent interactions involving non-catalytic domains, often termed adaptors, that may further interface with covalent modifications of proteins by phosphorylation and ubiquitination (*Lee and Peter, 2003*; *Kumar and Colussi, 1999*). These regulatory processes controlling the ultimate unleashing of the effector often occur in the context of large macromolecular complexes such as the apoptosome (*Riedl and Salvesen, 2007*).

Studies by us and others revealed that despite the protean manifestations of apoptosis and related phenomena in immunity across diverse eukaryotic lineages, at the molecular level it features certain common protein domains mediating different steps of the process (*Aravind et al., 2001*; *Aravind et al., 1999*; *Koonin and Aravind, 2002*). On the regulatory side, these include the NTP-dependent switches involved in the formation of the apoptosome, inflammosome, and related complexes in animals, fungi, and plants in the form of the STAND NTPases (e.g., AP-ATPases, NACHT, and DAP-3; *Figure 1B*) and AP-GTPases (*Leipe et al., 2004*). These NTPase domains are typically coupled in the

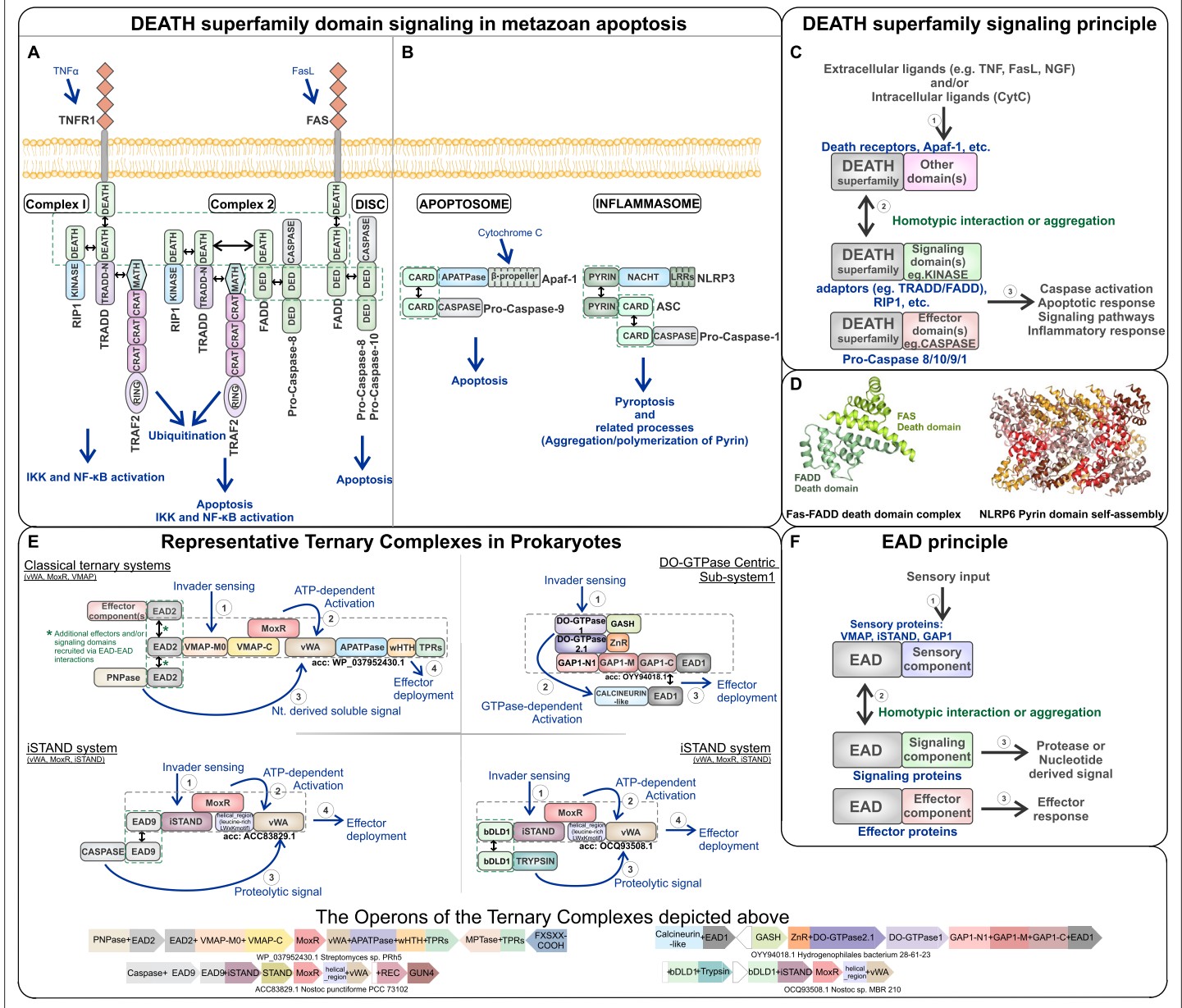

**Figure 1.** Signaling mechanism parallels between eukaryotic Death-superfamily domains and prokaryotic effector-associated domains (EADs) in their biological contexts. (**A**) Extrinsic and (**B**) intrinsic pathways in eukaryotic apoptosis signaling. (**C**) Schematic representation of the interactions mediated by Death domains in various metazoan signaling processes. (**D**) Cartoon structural representation of the Death-Death interaction in the FADD-FAS complex (PDB: 3EQZ) and the Pyrin domain-based protein-templated assembly of filamentous polymeric complexes in NLRP6 (PDB: 6NCV). (**E**) Representative ternary biological conflict systems where the EADs, predicted to perform roles comparable to eukaryotic Death domains, were discovered. (**F**) Schematic representation of the interactions mediated by the EADs in prokaryotic biological conflict systems that are predicted to lead to a highly regulated counter-invader response.

same polypeptide to supersecondary-structure-forming repeats, such as the LRR, WD, and TPR that provide a platform for the assembly of macromolecular complexes and the recognition of cell-death stimulating molecules (*Figure 1B*; *Leipe et al., 2004*; *Inohara and Nunez, 2001*). On the effector side, the domains shared by disparate apoptotic systems feature a common set of executors or toxin domains in the form of the caspase-like peptidase (caspase, paracaspase and metacaspase, and their relatives; hereinafter caspase), NucA/EndoG-like endonuclease, TIR, and ART domains (*Uren et al., 2000*; *Sanmartín et al., 2005*; *Schäfer et al., 2004*; *Narayanan and Park, 2015*; *Hassa et al., 2006*). Further, these studies also indicated that both the regulator and effector domains found across

eukaryotic apoptotic systems have their ultimate origins in prokaryotes, where they show a preponderant presence in multicellular forms (*Aravind et al., 1999*; *Koonin and Aravind, 2002*; *Kaur et al., 2020*).

In eukaryotes, these domains are commonly incorporated into lineage-specific protein domain architectures and embedded within signaling webs that are typical of eukaryotic regulatory systems, such as protein kinase cascades (e.g., MAP kinases) or the ubiquitin/ubiquitin-like (Ub/Ubl)-proteasome system (*Figure 1A*; *Aravind et al., 2001*; *Orlowski, 1999*; *Keshet and Seger, 2010*). Until recently, potential regulatory networks in which the prokaryotic homologs of apoptotic proteins were situated remained unclear. As part of our program to comprehensively identify new molecular components mediating biological conflicts (*Zhang et al., 2012*; *Iyer et al., 2017*; *Anantharaman and Aravind, 2003*; *Aravind et al., 2012*; *Burroughs et al., 2015*; *Zhang et al., 2016*; *Burroughs and Aravind, 2020*), we uncovered a class of thematically unified systems in phylogenetically distant prokaryotes with a predominantly multicellular habit (e.g., actinomycetes, cyanobacteria, chloroflexi, and myxobacteria; *Figure 1E*; *Kaur et al., 2020*). These possess counterparts of eukaryotic apoptotic protein domains as part of multidomain proteins predicted to form multimeric signaling complexes. These complexes are often characterized by a 'ternary' form, that is, with three functional categories of components predicted: (1) to detect invaders, (2) signal their presence and set a threshold for effector activation, and (3) finally unleash the effectors (*Figure 1E*). Notably, they show threshold-setting components that are based on chaperone-cochaperone pairs, proteolysis, nucleotide signals, and/or GTPase switches (belonging to the broader septin/GIMAP-like clade) that present analogies to the tight regulation seen in eukaryotic apoptotic systems (*Figure 1C–E*; *Kaur et al., 2020*). Selection for domain architectural and sequence diversification in these systems, especially in terms of their effector and invader-sensing components, suggested that the apoptosis-like processes mediated by these systems are part of the counter-invader response in multicellular prokaryotes.

These prokaryotic systems also brought to light a hitherto unappreciated commonality with metazoan apoptotic systems, namely the use of small 'adaptor' domains in complex formation and effector recruitment (*Figure 1F*). In metazoa, the regulators and effectors are brought together by the interactions of the adaptor domains, which are most commonly the α-helical domains of the Death-like superfamily, viz., the Death domain, Death effector domain (DED), caspase recruitment domain (CARD), and Pyrin domain (PYD) that have a shared six-helical bundle fold (*Park et al., 2007*; *Figure 1C*). For example, the C-terminal Death domains of the vertebrate tumor necrosis factor receptor (TNFR1), FAS, or the neurotrophin receptor p75, in response to activation by their respective ligands (TNFα, Fas-Ligand, and the nerve growth factor), recruit TRADD with a C-terminal Death domain, which in turn recruits the protein kinase Rip1 and FADD that also possess Death domains (*Micheau and Tschopp, 2003*; *Hsu et al., 1995*; *Stanger et al., 1995*; *Boldin et al., 1995*; *Yazidi-Belkoura et al., 2003*). FADD additionally possesses a DED, which in turn recruits caspases with DEDs to induce a proteolytic cascade, leading to apoptosis (*Figure 1A*; *Chinnaiyan et al., 1995*). The N-terminal region of TRADD contains a domain (TRADD-N; Pfam PF09034) structurally related to the small-molecule-binding ACT domain (RRM-like fold) (*Aravind and Koonin, 1999b*; *Park et al., 2000*), which recruits the MATH domain of the TNFR-associated factor 2 (TRAF2), leading to activation of JNK/AP1-kinase and inflammatory response pathways transcriptionally controlled by nuclear factor (NF)-κB (*Park et al., 2000*; *Hsu et al., 1996*; *Tsao et al., 2000*), thus uniting apoptotic and immune signaling (*Figure 1A*). Our recent work showed that both α-helical domains (including the first examples of domains distantly related to the metazoan Death-like superfamily) and those related to the TRADD-N are present as potential adaptors in the newly described conflict systems enriched in multicellular bacteria (*Kaur et al., 2020*).

These findings provided the first hints for the possible provenance of key apoptotic adaptors related to the Death-like and TRADD-N families in bacterial conflict systems. In this study, we expand these findings to show that the classical metazoan-type Death domains are found in bacterial conflict systems that possess a similar array of effectors as the metazoan apoptotic systems. We also identify several previously unrecognized families of the TRADD-N domain in bacteria and metazoans. The new metazoan TRADD-N families often show lineage-specific expansions (LSEs) with a remarkable diversity of domain architectures paralleling their contextual connections in prokaryotic conflict systems. Consequently, we show that the TRADD-N domains define a widespread, conserved functional principle for

regulating effector deployment across diverse conflict systems. These observations helped us identify and explain the mechanisms of several novel components of prokaryotic and animal immunity.

## Results

### The 'EAD principle' helps identify bacterial Death-like domains closely related to metazoan Death domains

Despite sharing a basic ternary organization of invader sensing, threshold setting, and effector deployment components, the recently identified conflict systems enriched in multicellular prokaryotes differ in terms of the actual components that play these roles (*Figure 1E*; *Kaur et al., 2020*). One major class of these have a threshold-setting regulatory core comprising a MoxR-type AAA+-AT-Pase-von Willebrand factor A (vWA) domain chaperone-cochaperone pair, with the effector domains typically occurring in the C-terminal region of the same polypeptide as the vWA domain (*Figure 1E*). Different versions of these systems are defined by one of several distinct third components, such as the vWA-MoxR-associated protein (VMAP; *Figure 1E*, top-left panel) or the inactive STAND NTPase module (iSTAND; *Figure 1E*, middle two panels). The system-specific third components have invader-recognition and multimeric assembly domains further linked to N-terminal signaling domains that are predicted to facilitate effector activation through nucleotide-derived or proteolytic signals. Other ternary systems have a core GTPase-switch along with their own invader recognition and effector components (*Figure 1E*, top-right panel). Despite these disparate elements, we observed that these systems frequently share one or more of 12 distinct, small, non-catalytic domains showing parallel domain-architectural and predicted functional features. Since these domains are typically coupled with effector domains, we termed them the effector-associated domains (EADs) (*Kaur et al., 2020*). In the ternary systems, they are most frequently coupled to the system-specific third component, for example, at the N-terminus of the VMAP or iSTAND proteins (*Figure 1E, F*). Additional genes encoding EAD proteins occur in the same operon as the genes for the core ternary system or elsewhere in the genomes of organisms possessing such systems (*Figure 1E*, lower panel). Further, the EADs encoded in genomic proximity to each other or by the same organism tend to be more closely related than their counterparts from other genomes (*Kaur et al., 2020*).

Thus, by tracking the EADs we were able to identify other predicted core counter-invader conflict systems beyond the original ternary systems. Their domain compositions pointed to diverse activating mechanisms, such as proteolysis by either trypsin-like or caspase-like peptidases (respectively NucA-trypsin and EACC1 systems; *Kaur et al., 2020*). Together, these observations helped define a widespread organizational feature of these conflict systems, that is, the EAD principle: the EADs increase the range of effector and signaling domains that can be recruited to and deployed by a core system via homotypic interactions (*Figure 1*; *Kaur et al., 2020*). This pointed to a functional analogy between the EADs in these bacterial systems and the homotypic interactions mediated by the Death-superfamily adaptors in metazoan apoptosis/immune systems (*Figure 1C and F*; *Park et al., 2007*). Augmenting this equivalence, we found that 9 out of the 12 EADs are α-helical domains with structural parallels to the metazoan Death-like superfamily domains (*Kaur et al., 2020*; *Supplementary file 1*). Moreover, we found two distant bacterial homologs of the Death-like superfamily among the EADs, namely bDLD1 (EAD3) and bDLD2. These are present in the vWA-MoxR ternary systems respectively with VMAP and iSTAND modules as their third component (*Kaur et al., 2020*).

Instigated by these findings, we used the procedure of domain architectural analysis followed by iterative sequence profile searches to find any new EADs that might throw further light on their evolutionary and/or functional relationship to the Death-like superfamily. As a result, we retrieved a novel family of bacterial Death-like domains (e.g., GenBank accession: OUL31312, *Nostoc* sp. T09; *Figure 2*) associated with predicted counter-invader systems such as the vWA-MoxR-iSTAND ternary system (*Figure 2A, B, C and D*). We accordingly named this the bacterial Death-like domain-3 (bDLD3). Remarkably, unlike bDLD1 and bDLD2, bDLD3 is closely related to metazoan Death domains (*Figure 2D*). For instance, sequence profile searches with a bDLD3 domain from *Nostoc* (GenBank: OUL31312.1) as query recovered the eukaryotic Death domain in iterative PSI-BLAST searches (e.g., the nematode *Gongylonema* VDN25015.1 in iteration 2, e-value $10^{-6}$). A multiple alignment of bDLD3 showed that it shares key conserved residues with the classical metazoan Death domain including tryptophans in helices 2 and 4, and an RxD motif at the beginning of the terminal helix 6 (*Yang*

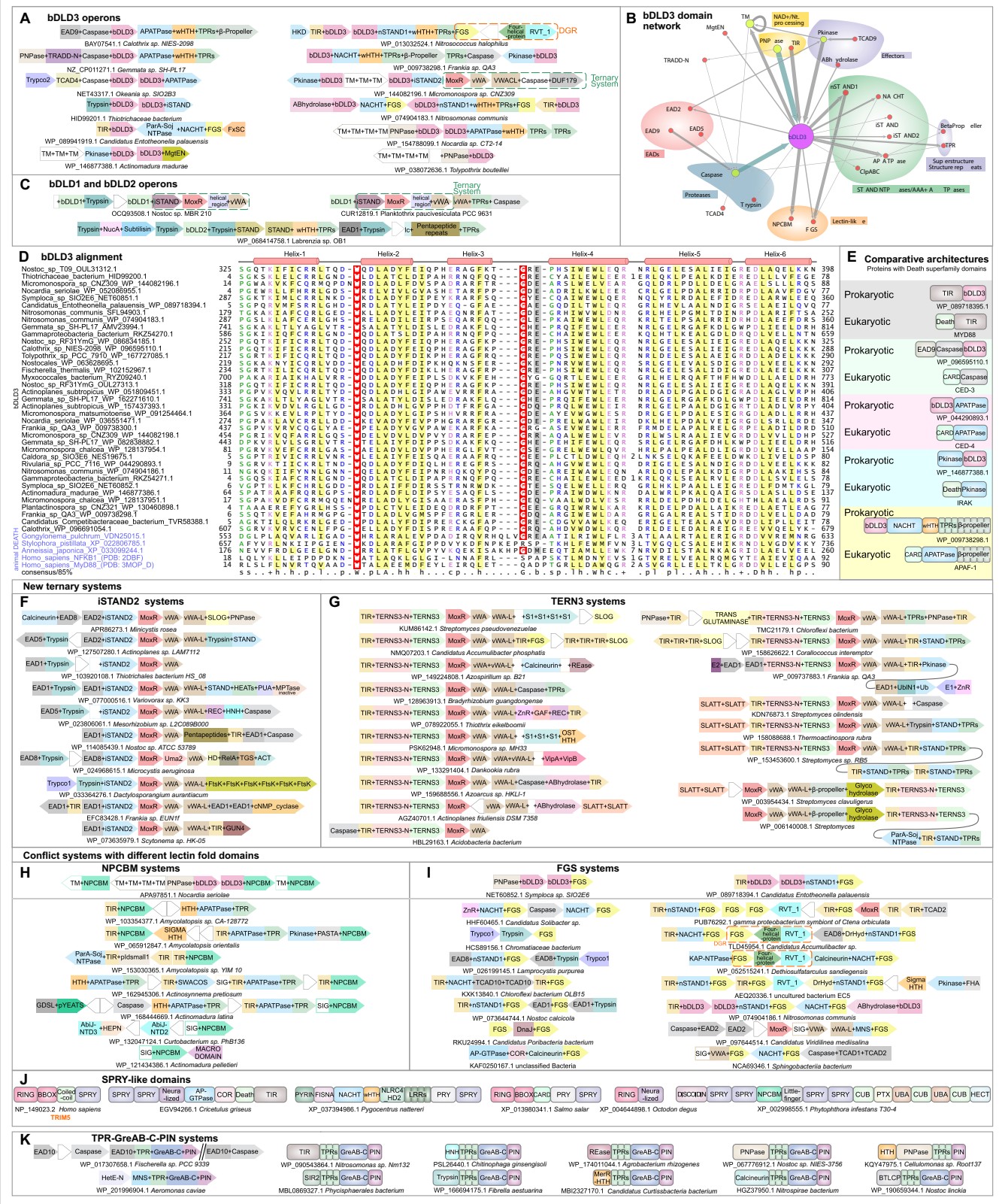

**Figure 2.** Domain and gene neighborhood contexts of the bacterial Death-like domains (bDLD). (**A**) Representative gene neighborhoods coding for bDLD3 proteins. (**B**) Domains architectural network of bDLD3. (**C**) Representative gene neighborhoods coding for bDLD1 and bDLD2 proteins. (**D**) Multiple sequence alignment (MSA) of bDLD3 and representative eukaryotic Death domains (in purple). Sequences are denoted by the organism name and NCBI protein accession number separated by an underscore. Domain limits are numbered. . The predicted secondary structure and the sequence

*Figure 2 continued on next page*

Figure 2 continued

consensus at 85% identity are depicted above and below the alignment, respectively. Coloring is as per the consensus abbreviation of residue type, where s: small; u: tiny; +: basic; h: hydrophobic; l: aliphatic; p: polar; b: big. In all figures, α-helices and β-strands in MSAs are depicted as cylinders and arrows, respectively. Insertions in the sequences are represented by the number corresponding to their length. (E) Comparable domain architectures of the bDLD3 and Death-superfamily domains. iSTAND2 (F) and TERNS3 (G) containing novel ternary conflict systems. Novel conflict systems utilizing the lectin fold domains NPCBM (H) and FGS (I). (J) Domain architectures of eukaryotic SPRY-like domains. (K) Novel conflict system with a constant core comprising TPR, GreA/B-C-terminal, and PIN domains.

The online version of this article includes the following source data for figure 2:

**Source data 1.** Comprehensive gene neighborhoods and domain architectures of systems described in the figure.

**Source data 2.** Multiple sequence alignments of novel domains described in this study.

et al., 2005; Weber and Vincenz, 2001; Figure 2 and Figure 2—source data 2). bDLD3 is more widespread in bacteria and displays a broader range of domain architecture and gene neighborhood associations than bDLD1 and bDLD2 (Figure 2A and C). Like the earlier described ternary systems, bDLD3 shows a statistically significant propensity to occur in bacteria with a multicellular habit, such as members of actinobacteria, proteobacteria, cyanobacteria, nitrospinae/tectomicrobia, and planctomycetes (p=3.179 × 10⁻¹⁴; calculated using the hypergeometric distribution; see Materials and methods). Predicted effector domains fused to bDLD3 include catalytic domains involved in NAD⁺/nucleotide-processing (TIR, purine/pyrimidine nucleotide phosphorylase [hereinafter PNPase]) (Essuman et al., 2018; Mao et al., 1997), protein modification (S/T kinase) (Taylor and Kornev, 2011), proteolysis (caspase and trypsin) (Aravind and Koonin, 2002; Barrett and Rawlings, 1995), and hydrolysis of membrane-lipids (α/β-hydrolase) (Holmquist, 2000; Figure 2A, B). The versions fused to caspases are often linked in an operon to a gene encoding an AP-ATPase (a member of the STAND clade of NTPases typified by the animal apoptosome proteins APAF1/Ced-4; Figure 2E; Leipe et al., 2004; Saleh et al., 1999; Chaudhary et al., 1998). As with other EADs, bacteria (where complete genomes are available) frequently possess multiple copies of bDLD3, where one copy is fused to C-terminal effectors and the other to the N-termini of core components of the ternary system (Figure 2A). For example, mimicking the earlier-reported fusion in a ternary system of the bDLD1 to the iSTAND module (Kaur et al., 2020), in this work we recovered a fusion of bDLD3 to the iSTAND domain in a *Thiotrichaceae proteobacterium* (HID99200.1; Figure 2C). Again, like in the first system where an adjacent gene codes for a second copy of bDLD1 fused to a trypsin-like peptidase effector domain, in the current system we found a second neighboring gene coding for a bDLD3 domain fused to a trypsin-like peptidase domain (Figure 2C). The same organism also has two other bDLD3 domains elsewhere in the genome, of which one is fused to a Clp-like AAA+ ATPase (Hoskins et al., 2001).

## bDLD3 helps identify two new versions of MoxR-vWA ternary systems

We next used the gene neighborhood- and domain architecture-associations of bDLD3 in conjunction with sensitive sequence analysis as a gateway to explore the conflict systems that contain it (Figure 2—source data 1, see Materials and methods). Consequently, we detected systems in which the bDLD3-encoding gene was genomically linked to 3' MoxR-vWA pairs typical of other ternary systems (e.g., *Micromonospora*, WP_144082196); however, in these it was fused to an unknown module that took the place of the third component VMAP or iSTAND modules of the previously described ternary systems (Figure 2F). This unknown module was additionally seen in several MoxR-vWA systems, where it was fused to N-terminal EAD1 or EAD8 domains that took the place of bDLD3 (Figure 2F). Through profile-profile comparisons with the HHpred program, we found this unknown module to be a previously unrecognized inactive STAND NTPase module (p-value 10⁻⁶ for iSTAND and p=1.2–2.2 × 10⁻⁴ for other STAND NTPases); accordingly, we named it iSTAND2 (Figure 2F). In the known VMAP and iSTAND systems (Kaur et al., 2020), the effector domains are most often directly connected to the C-terminus of the vWA component by an α-helical linker domain, the vWA-L. In the iSTAND2 systems, as in a minority of iSTAND systems, the vWA-L with C-terminal effector domains is encoded by a separate gene adjacent to that coding for the vWA domain (Figure 2F). Using this gene neighborhood template, we identified a further class of MoxR-vWA ternary systems that contain a gene coding for a protein with the vWA-L fused to C-terminal effector domains adjacent to the separate vWA gene (Figure 2G). The conserved 5' gene of these systems codes for the third component typified

by a novel α + β domain distinct from the third component of all other MoxR-vWA ternary systems; we named it TERNS3 for ternary system component 3 (*Figure 2G*). Both these new systems, like the earlier-described ternary systems, are mainly found in multicellular cyanobacteria, actinobacteria, chloroflexi, and proteobacteria (iSTAND2: p=2.67 × 10⁻³; TERNS3: p=4.106 × 10⁻¹⁷; *Figure 2—source data 1*, *Supplementary file 2*).

Both iSTAND2 and TERNS3 are linked to some of the same signaling domains associated with the VMAP and iSTAND modules, either via direct N-terminal fusions or via fusions to EADs (*Figure 2F, G*; *Kaur et al., 2020*). In the case of iSTAND2, this linked domain is predominantly a trypsin-like domain and infrequently a cNMP cyclase domain (*Murzin, 1998*), which are fused to either the same EADs or bDLD3 corresponding to that found at the N-terminus of iSTAND2 (*Figure 2F, G*). In the TERNS3-containing systems, the most common association is with the TIR domain and less commonly with proteases (e.g., the circularly permuted transglutaminase-like peptidase; *Anantharaman et al., 2001*), both showing direct fusions. These contextual connections suggest subtle differences in the predominant regulatory mode of the iSTAND2 and TERNS3 ternary systems, with the former probably mainly depending on a proteolytic activation step and the latter typically using a NAD⁺-derived signal with a ADP-ribose (ADPr) moiety generated by the TIR domain (*Burroughs and Aravind, 2020*; *Essuman et al., 2018*).

In addition to the effectors fused to the vWA-L domain, some of these systems contain genes coding for additional effectors fused to the cognate EADs (*Figure 2F and G*). The classes of effector domains found in the iSTAND2- and TERNS3 systems overlap with those in the VMAP and iSTAND ternary systems (*Kaur et al., 2020*). These include restriction endonuclease fold (REase) (*Steczkiewicz et al., 2012*), supersecondary structure forming α-helical repeats, small-molecule kinase (related to aminoglycoside kinases) (*Hon et al., 1997*), S/T/Y protein kinases, TIR, FtsK ATPases (both active and inactive copies) (*Iyer et al., 2004*), and the cyanobacteria-specific tetrapyrrole-binding GUN4 and FGS domains (*Davison et al., 2005*; *Doulatov et al., 2004*; *Figure 2F,G*). Both iSTAND2- and TERNS3 systems, like the previously described VMAP-ternary systems, also feature receiver domains occurring independently of histidine kinases as potential effectors. We predict a role for these comparable to the recently described versions of the receiver domain in biological conflict systems that might function independently of histidine kinases as nucleotide-responsive switches (*Burroughs and Aravind, 2020*; *West and Stock, 2001*; *Pao and Saier, 1995*; *Gao et al., 2019*; *Iyer et al., 2021*). Unique to the iSTAND2 systems are effectors containing a modified nucleic-acid-binding PUA domain fused to a STAND NTPase and inactive zincin-like metalloprotease domains (e.g., *Varivorax*; WP_077000516.1; *Figure 2F*; *Iyer et al., 2006a*; *Stöcker et al., 1995*). Unique to certain TERNS3 systems are effector modules with the RNA-binding OST-HTH domain fused to multiple OB fold S1 domains (*Anantharaman et al., 2010*; *Figure 2G*). These point to the potential recognition of both modified and unmodified nucleic acids as part of the response to the invasive elements by these systems. In some TERNS3 systems, we observed a unique signaling ensemble in the form of a Ubl-conjugation system (*Iyer et al., 2006b*; *Burroughs et al., 2012a*) with an E1 ligase coupled to Ubl and E2 components, both fused to EAD1s (*Figure 2G*). These are likely recruited to the core TERNS3 domain that is furnished with its own N-terminal EAD1.

## bDLD3 and EAD10 help identify counter-invader systems respectively with lectin fold and RNA polymerase-association modules

EADs link biochemically disparate effector domains to the regulatory core of counter-invader conflict systems. These effector domains might also be found independently of EADs as part of other (often simpler) counter-invader systems. Thus, the EAD-linked domains help define potential novel effectors and conflict systems that possess them. This postulate helped us identify multiple previously unrecognized sets of conflict systems, two of which utilized lectin fold domains (*Figure 2H,I*) and were uncovered via bDLD3, and another one predicted to associate with the RNA polymerase (RNAP) that was discovered via EAD10.

The first of these are typified by the NPCBM (novel putative carbohydrate binding module; Pfam: PF08305) (*Rigden, 2005*), a member of the discoidin-like fold that includes numerous carbohydrate-binding (lectin) domains (*Baumgartner et al., 1998*). We observed loci with tandem genes (e.g., WP_052086955.1 of *Nocardia seriolae*) specifying proteins wherein bDLD3 is fused to either NPCBM or PNPase domains (*Figure 2H*, top panel). Closely related NPCBM domains are frequently fused to

known effectors such as TIR and S/T/Y-type kinases and are encoded in conserved gene neighborhoods coding for an AP-ATPase fused to TPR repeats and caspases (*Figure 2H*, bottom panel). The second of these systems are centered on the FGS domain (the so-called 'Formylglycine-generating enzyme sulfatase' in Pfam: PF03781) (*Alayyoubi et al., 2013*), which has the same protein fold as another large class of lectin domains, the C-type lectins (*Figure 2I*; *McMahon et al., 2005*). In the cyanobacterium *Symploca* (NET60851.1), we found bDLD3 in an identical configuration as the above-described fusion with the NPCBM domain, except that the latter domain was replaced by the FGS domain (*Figure 2I*, top panel). Further, a conserved gene neighborhood found in certain proteobacteria and nitrospinae codes for a pair of bDLD3 proteins: one of the bDLD3 domains is fused to STAND NTPase and FGS domains (acc: WP_089718394.1) and the other to a TIR domain (*Figure 2I*). This similarity in the domain architectural and contextual connections of NPCBM and FGS domains suggests that these are indeed functionally comparable.

The above-noted STAND NTPase-FGS combination is also found widely across several bacterial and archaeal lineages independently of its association with bDLD3 (*Figure 2I*). In these cases, the N-terminal bDLD3 is often displaced by a slew of other EADs (EAD1, EAD2, EAD7, and EAD8) or effector domains belonging to diverse functional classes: for example, nucleotide and NAD$^+$-processing domains (TIR, DRHyd, calcineurin), peptidases (caspase, trypsin), and protein phosphorylation-related domains (S/T/Y-kinase, FHA) (*Burroughs et al., 2015*; *Aravind and Koonin, 1998a*; *Durocher and Jackson, 2002*). The central NTPase also shows some diversity – it might either belong to the NACHT clade or the MNS clade (Npun2340/2341 family) (*Leipe et al., 2004*) or a novel STAND NTPase clade that we term nSTAND1 (sometimes also present in the previously described EACC1 conflict systems; *Kaur et al., 2020*; *Figure 2I*). The STAND NTPase domain might also be displaced by other types of NTPase domains, namely the KAP ATPase (*Aravind et al., 2004*), which was previously described as a player in anti-bacteriophage immunity (*Clark et al., 2014*) or the AP-GTPase, a small GTPase domain of the Ras-like clade (*Figure 2I*), which is found in animal apoptotic proteins (e.g., the DAP protein kinase [*Kawai et al., 1999*]). As in the ternary conflict systems, the EAD genes associated with these systems are typically found in multiple copies. Here again, one copy of the EAD is fused to the N-terminus of aforesaid NTPases and the rest to distinct effectors (*Figure 2H,I*). We had earlier noted the FGS domain as an effector in several ternary conflict systems (*Kaur et al., 2020*). As in those systems, here too the FGS genes are often associated with the components of the diversity generating system (DGR) (*Wu et al., 2018*), namely a reverse transcriptase and the four-helical domain accessory protein (*Figure 2I*). These occur as either a neighboring locus or elsewhere in the genome. Hence, it is likely that as in the classical DGR and ternary systems, the FGS domain is diversified by the error-prone action of the reverse transcriptase (*Alayyoubi et al., 2013*). Further, organisms often have two or more copies of the NTPase-FGS system, each with distinct NTPase types and N-terminal EADs or effectors suggesting selection for diversification to cover for resistance on the side of the invaders.

The recruitment of two distinct lectin fold domains, NPCBM and the FGS, to conflict systems, along with their similar contextual connections (*Figures 2H and 1*) is reminiscent of the SPRY-like domains (SPRY, PRY, and Neuralized repeat) with the concanavalin lectin fold (*Ponting et al., 1997*) that are prominent in eukaryotic immunity (*D'Cruz et al., 2013*). In eukaryotes, the SPRY-like domains are coupled to effectors such as the RING finger or HECT domain E3 Ub-ligase (the TRIM proteins) (*Reymond et al., 2001*), or the NACHT clade of STAND NTPases or different members of the Death-like superfamily (e.g., Death, Pyrin, and CARD) (*Papin et al., 2007*; *Chae et al., 2006*; *Figure 2J*). In several stramenopiles, the above-mentioned Ub-ligases might additionally possess a NPCBM domain (*Figure 2J*). In the well-studied TRIM systems, SPRY domains directly recognize viral molecules (*Perez-Caballero et al., 2005*; *James et al., 2007*) (e.g., the glycosylated HIV capsid protein [*Yap et al., 2005*]). Based on these parallels, we propose that the NPCBM and the FGS domains directly interact with invasive molecules (viral nucleic acids or proteins) and recruit other catalytic effectors to evince downstream action, such as apoptosis, to limit the invader. Since lectin domains can recognize glycosylated macromolecules, these could be one possible target (see below). In the case of the FGS domain, the associated DGR suggests that these could be potential analogs of vertebrate adaptive immunity receptors that undergo hypermutation to bind a wide diversity of invader molecules (*Boehm et al., 2018*; *Flajnik and Kasahara, 2010*). Further, the significant preponderance of both NPCBM (actinobacteria) and FGS (cyanobacteria) systems in multicellular bacteria (p=2.7 × 10$^{-36}$ and p=3.7 × 10$^{-5}$, respectively) implies that they are part of the unique immune processes of such organisms, which

are regulated by the associated NTPase domains, ternary systems, or other fused signaling domains (*Kaur et al., 2020*).

In a similar vein, EAD10 led us to a previously unrecognized prokaryotic conflict system that is widespread in bacteria and sporadic in archaea. The archetypal member of this system features an EAD10 domain fused to a constant (but rapidly evolving) C-terminal region comprised of TPR repeats a GreA/B-C domain, and a hitherto undetected version of the PIN domain (WP_017307658.1 from the cyanobacterium *Fischerella* sp.; *Figure 2K*). In other examples of this system, EAD10 is replaced by 1 of 13 different effector domains, such as DNase (REase, HNH), NAD⁺/nucleotide-targeting (PNPase, TIR, SIR2), and peptidase (trypsin, papain fold transglutaminase-like domain: BTLCP) domains, and an uncharacterized effector domain also found at the N-terminus of certain fungal heteroincompatibility HetE proteins.

The GreA/B-C is a domain related to the FKBP-like peptidyl prolyl isomerases that occurs in the bacterial transcription elongation factors GreA and GreB (*Stebbins et al., 1995*) and transcription inhibitors like Gfh1 (*Tagami et al., 2010*). This domain engages the universally conserved coiled-coil segment that forms part of the RNAP secondary exit channel in the β' subunit, which is needed to accommodate the 3' end of the back-tracked transcript and the diffusion of soluble nucleotide substrates (*Abdelkareem et al., 2019*). Given that these systems mostly occur as standalone genes, with few conserved gene neighbors, we predict that they act by themselves on the RNAP complexes that have been hijacked to transcribe viral DNA. It is conceivable that the GreA/B-C domain of these proteins engages the RNAP β' coil-coil as in conventional transcription elongation and causes the transcript to be extruded into the secondary channel. It can then be cleaved by the C-terminal PIN domain. However, if this action on the transcript were to fail due to viral inhibition or the overwhelming of this line of defense, the N-terminal variable effector could be activated either directly or via EAD-EAD interaction to unleash an apoptotic response.

## Bacterial and metazoan immunity systems with a vast array of TRADD-N domains

TRADD is a versatile adaptor protein that interacts via its Death domain with the cytoplasmic Death domains of multiple members of the TNFR family, activating different apoptotic pathways (*Micheau and Tschopp, 2003*; *Hsu et al., 1995*). It recruits the Ub E3 ligase TRAF2 through an interaction between its N-terminal domain (TRADD-N) and the β-sandwich MATH domain of the TRAF (*Hsu et al., 1996*; *Figure 1A*). The TRADD-N domain was until recently only known from vertebrates. We recently reported bacterial TRADD-N domains that were recovered via their fusion to EAD1 and EAD4 (*Kaur et al., 2020*). In the current study, analysis of the bDLD3 proteins led us to another bacterial homolog of the TRADD-N domain with fusions to an N-terminal PNPase and C-terminal caspase (e.g., AMV23994 from *Gemmata* sp. SH-PL17; *Figure 3A*, first architecture). In a closely related organism *Gemmata massiliana,* the TRADD-N domain is instead fused to N-terminal nucleotide cyclase and FGS domains and a C-terminal caspase domain (*Figure 3A*, second architecture). The fusion to effector domains or EADs and the diversity of architectures in closely related species supports a role for these TRADD-N proteins in counter-invader conflicts. Accordingly, we systematically investigated these TRADD-N domains to identify new homologs.

Using bacterial TRADD-N homologs as queries in PSI-BLAST searches against the nr50 and the nr90 databases (see Materials and methods), we recovered novel TRADD-N families across several bacteria and animals. For example, PSI-BLAST searches initiated with a TRADD-N domain of *Anaerolineales bacterium* (GenBank: MBE0669634.1, region 141.232), in addition to numerous bacterial sequences, recovered several metazoan sequences with significant scores (e = $10^{-5}$-$10^{-12}$ within five iterations; e.g., amphioxus XP_035660412.1 e = $10^{-8}$ in iteration 4). This search also recovered borderline hits to two starfish proteins (XP_022106408.1; *Acanthaster planci*; region 58.150, e = .08; XP_033636006.1, *Asterias rubens*; region 300.388, e = 0.09, iteration 6). Their relationship to the bacterial proteins was confirmed by reciprocal searches: for instance, a PSI-BLAST search seeded with the above region from the *A. planci* recovered significant hits to bacteria homologs (e.g., NEP55853.1 from the cyanobacterium *Symploca*, e = $10^{-5}$). Also recovered in these searches were the DNA-binding Death effector domain-2 (DEDD2/FLAME3) proteins of metazoans. Further, a profile-profile search initiated with an alignment generated from these echinoderm sequences and their bacterial homologs unified them with the vertebrate TRADD-N domains (HHpred probability 95.45%), thereby confirming them as TRADD-N domains.

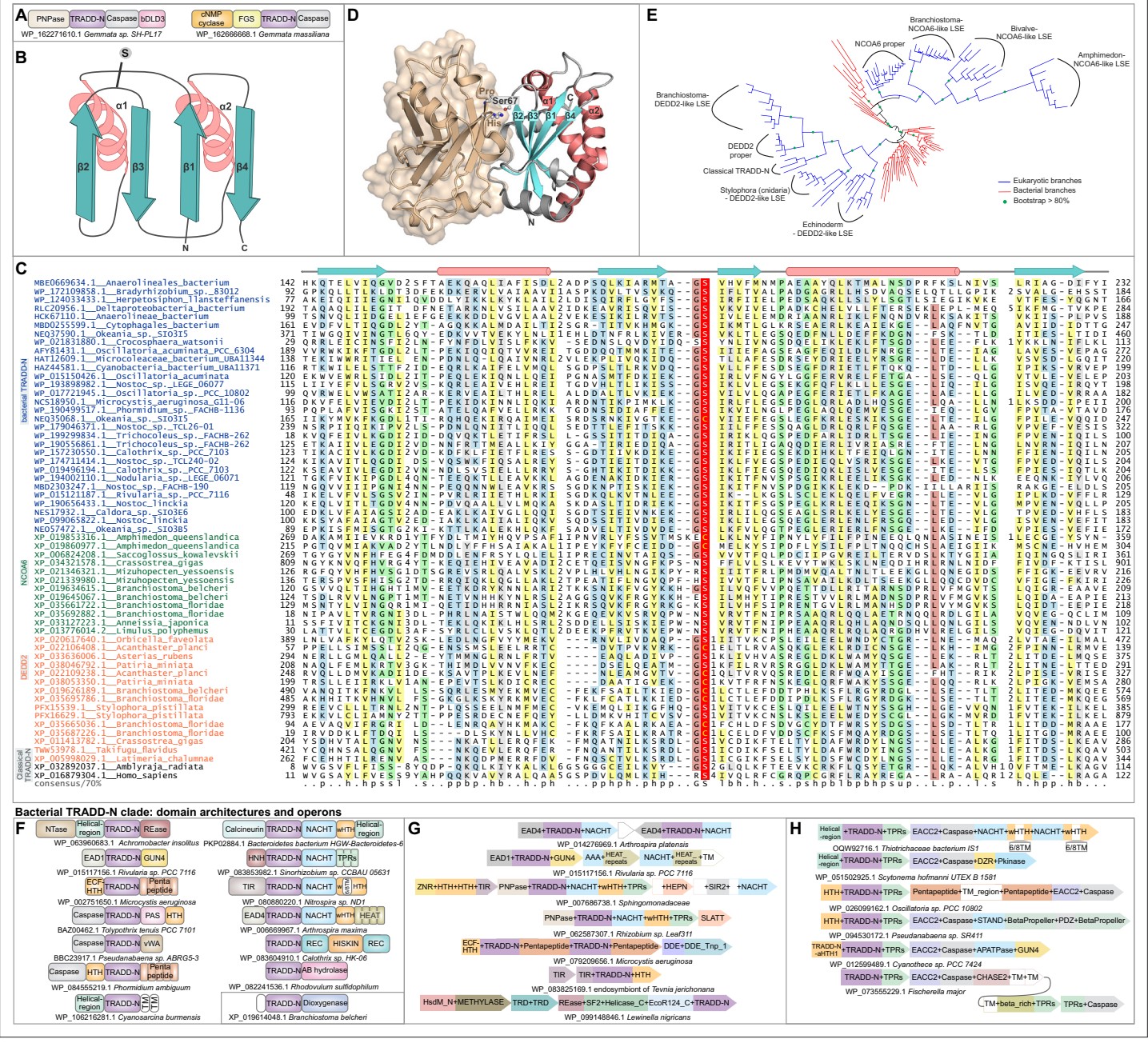

**Figure 3.** Structure, alignment, phylogeny, and contextual analysis of the TRADD-N domain. (**A**) The bacterial TRADD-N domains that were initially recovered in searches. (**B**) Topology diagram of TRADD-N. Arrows and helices represent β-strand and α-helical regions, respectively. The conserved serine residue at the β2-β3 turn is indicated. (**C**) Multiple sequence alignment (MSA) of TRADD-N. Refer to *Figure 2* legend for details of the MSA rendering. (**D**) The interaction of TRADD-N with the MATH domain (PDB: 1F3V). Residues mediating the non-covalent interactions are indicated. (**E**) Phylogenetic tree of representative TRADD-N domains showing the major clades . Domain architectures of (**F**) and gene neighborhoods coding for (**G**, **H**) bacterial TRADD-N domain proteins.

The online version of this article includes the following source data and figure supplement(s) for figure 3:

**Source data 1.** Comprehensive gene neighborhoods and domain architectures of the bacterial TRADD-N domains.

**Figure supplement 1.** TRADD-N phylogenetic tree showing the names of the branches.

Among the hits recovered in the first of the above-mentioned searches were the animal nuclear receptor coactivator 6 (NCOA6, e.g., XP_033127223.1 from the feather star [Echinodermata] *Anneissia japonica*, e-value: $10^{-5}$, iteration 5). This hit overlapped with a model termed 'Nucleic_acid_bd' in the Pfam database (PF13820) defined by animal-specific proteins homologous to NCOA6. Their relationship was confirmed by a profile-profile search with the TRADD-N domain from the cyanobacterium *Nostoc* (PHJ73282.1, region 103.194) that also recovered the same NCOA6-derived Pfam profile (HHpred probability 98.2%). A multiple sequence alignment helped define the correct boundaries of this conserved domain in the animal NCOA6 proteins and secondary structure prediction along with a sampling of the conserved sequence motifs (see below) showed a complete congruence to the TRADD-N domain structure. Further, profile-profile searches with representatives of each of the newly recovered groups also hit versions of the conventional ACT and structurally related domains (e.g., Dystroglycan DAG1 and acylphosphatases; HHpred probability 68–72%) consistent with the previously established shared presence of a RRM-like fold in these domains and TRADD-N (*Figure 3B and C*; *Park et al., 2000*; *Tsao et al., 2000*). Additional transitive searches with these newly detected sequences recovered vast expansions of TRADD-N domains in several animals, such as the sponge *Amphimedon*, starfishes and the amphioxus *Branchiostoma*, the human DEDD2 protein, and its vertebrate homologs (*Figure 4—source data 1*).

The RRM-like fold of the TRADD-N domain is a two-layered α + β sandwich characterized by the kinking of its two α-helices and the exposed face of its four-stranded antiparallel β-sheet that forms a prominent interaction surface (*Figure 3B*; *Park et al., 2000*; *Tsao et al., 2000*). A comprehensive multiple sequence alignment revealed a strongly conserved uS motif (where u is a tiny residue, usually glycine) in the turn between β2 and β3, with the serine residue occasionally substituted by a cysteine (*Figure 3C*, *Figure 2—source data 2*). The crystal structure of the TRADD-N-TRAF2-MATH complex (PDB: 1F3V) shows that the corresponding Ser67 is a key determinant of the interaction of the TRADD-N domain with a proline in its partner, the MATH domain (*Figure 3D*; *Park et al., 2000*). The conservation of this residue suggests that it is part of a conserved interaction interface across the newly defined TRADD-N superfamily. We used a comprehensive multiple sequence alignment of the recovered TRADD-N domains to construct a phylogenetic tree (*Figure 3E*). Despite being a small domain, it showed multiple well-separated metazoan LSEs within them. The bacterial sequences appear to form a basal group from within which two distinct metazoan clade clades have emerged. The first of these encompasses the NCOA6 and related LSEs of metazoan proteins. The second encompasses the DEDD2, vertebrate TRADD, and related LSEs from various metazoa (*Figure 3E*). We describe each of the clades below in greater detail.

## The bacterial TRADD-N proteins

The bacterial homologs of the TRADD-N domain are significantly over-represented in bacteria showing multicellular habit or complex developmental cycles, namely cyanobacteria, certain proteobacteria, bacteroidetes, nitrospirae, planctomycetes, and actinobacteria ($p=1.942 \times 10^{-40}$). The bacterial TRADD-N domains are found in multidomain proteins, usually occurring in-between distinct N- and C-terminal domains. The N-terminal domains tend to vary greatly even between closely related organisms (*Figure 3—source data 1*, *Figure 3F*) and include enzymatic ones such as TIR, PNPase, HNH, REase, calcineurin-like phosphoesterase, caspase, or a novel nucleotidyltransferase domain of the DNA polymerase-β superfamily (*Aravind and Koonin, 1999a*) or non-enzymatic domains, viz., extra-cytoplasmic sigma-factor (ECF)-like HTH (*Heimann, 2002*) or EADs (EAD1, EAD4, EAD7). The overlap in these domains with effector domains and EADs found in other counter-invader conflict systems suggests that they are the likely effector components of these systems (*Figure 3F*). However, the ECF-like HTH domain points to a transcriptional response as a unique aspect of some of these systems. The C-terminal region typically contains supersecondary-structure-forming repeats such as pentapeptide, TPR, and HEAT (*Das et al., 1998*; *Groves et al., 1999*; *Bateman et al., 1998*), or other enzymatic (STAND NTPase, caspase), small-molecule-binding (PAS [*Ponting and Aravind, 1997*], GUN4), adaptor (bDLD3), or DNA-binding (HTH) globular domains (*Aravind et al., 2005*; *Figure 3F*). Some of these TRADD-N proteins are encoded in conserved gene neighborhoods with a further gene coding for a STAND NTPase of the AP-ATPase clade fused to C-terminal TPR repeats (*Figure 3G*).

Another group of the bacterial TRADD-N proteins are also encoded in conserved two-gene neighborhoods (*Figure 3H*), whose second gene codes for a protein with a constant N-terminal module

comprised of a novel α + β domain followed by a caspase domain. We termed the former domain EACC2 (Effector-associated Constant Component 2) by analogy to the similarly organized, previously described EACC1 systems with a comparable constant component (*Kaur et al., 2020*). The C-terminus of this protein is highly variable and contains one of several domains that might be either extracellular with accompanying TM helices (e.g., CHASE2 – a small-molecule receptor domain; *Zhulin et al., 2003*) or intracellular (e.g., PDZ, *Doyle et al., 1996*, S/T/Y-type protein kinase, and STAND NTPase domains; *Figure 3H*). Versions of this second gene also occur independently of the TRADD-N component, and in those instances the C-terminal region is again highly variable with fusions to several small-molecule-binding domains, supersecondary forming repeats, ADP-ribose-processing macro domains (*Slade et al., 2011*; *Peterson et al., 2011*), trypsin-like peptidases, and multi-TM domains that might form membrane pores (*Figure 3H*). In the majority of these neighborhoods that are independent of the TRADD-N gene, there are additional genes encoding an ECF sigma factor and a TPR protein (Pfam: DUF1822). Given that the organization of these systems with the constant EACC2 and caspase domains is reminiscent of the previously described EACC1 systems (*Kaur et al., 2020*), we propose that these proteins might sense stimuli that then induce an autopeptidase activation of the system via the action of the constant caspase domain. The organization of these systems indicates that the sensory aspect of this system would involve an interplay with the C-terminal variable domains, and an additional ultimate step involving the TRADD-N gene product and/or the coupled ECF sigma factor that can mediate the transcription regulation of a multiplicity of genes.

The closest eukaryotic homologs of the bacterial TRADD-N domains are found in most metazoan lineages except the vertebrates. These show a conserved N-terminal extension with a predicted caspase-cleavage site in the form of a DEhD motif (h is a hydrophobic residue in 80% of the sequences) (*Song et al., 2012*). Additionally, these TRADD-N domains are fused to a C-terminal 'FAM124 domain' (Pfam: PF15067; *Figure 3F*, inset; see below).

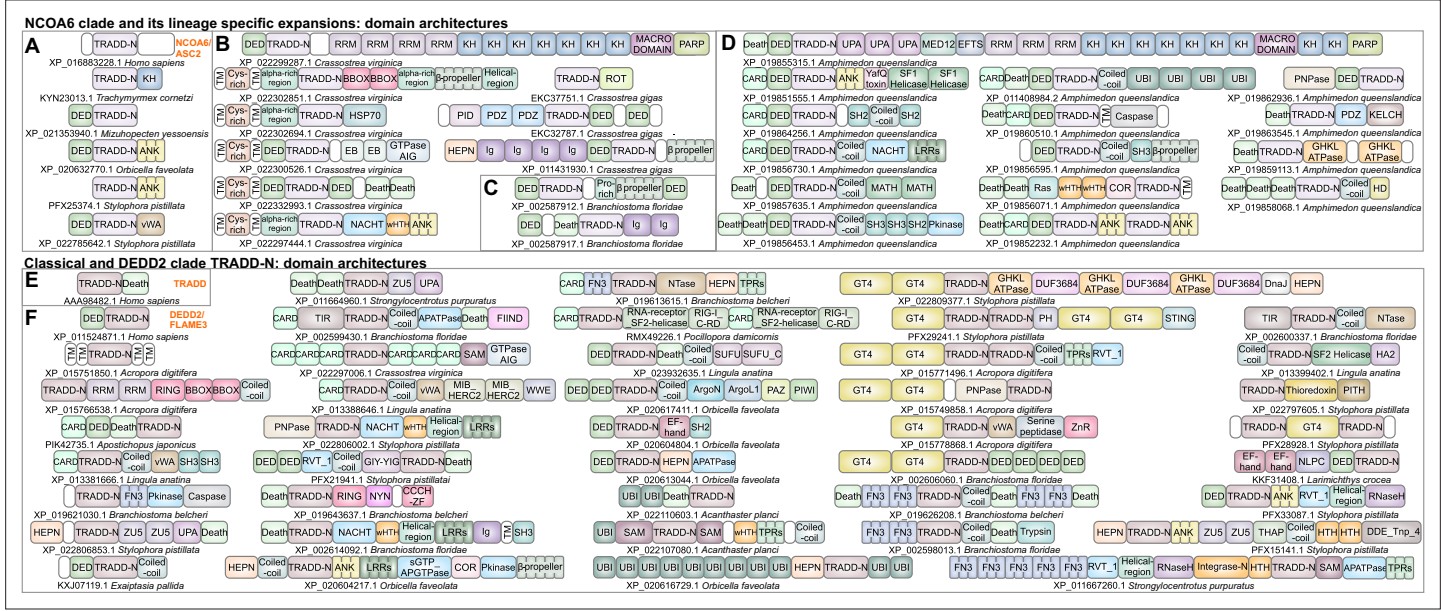

**Figure 4.** Domain architectures of NCOA6, classical TRADD-N, and DEDD2-like TRADD-N domains. (**A**) Domainarchitectures of representative TRADD-N proteins from the NCOA6 clade. Lineage-specific domain architectures of NCOA6 clade TRADD-N proteins from (**B**) *Crassostrea*, (**C**) *Branchiostoma*, and (**D**) *Amphimedon*. (**E**) Domain architecture of the classical TRADD-N protein. (**F**) Domain architectures of representative DEDD2 TRADD-N proteins.

The online version of this article includes the following source data for figure 4:

**Source data 1.** Eukaryotic TRADD-N domain architectures and phyletic distribution.

**Source data 2.** Non-redundant counts of TRADD-N domains in eukaryotic lineages.

## NCOA6 clade

In this study, we identified a previously unrecognized version of the TRADD-N domain in the metazoan transcription regulator NCOA6 (also known as ASC-2/AIB3/NRC/PRIP/RAP250/TRBP), a component of SET domain histone methyltransferase complexes such as the ASCOM coactivator, involved in nuclear hormone receptor transactivation (*Goo et al., 2003*; *Lee et al., 2009b*; *Lee et al., 2009a*; *Hillmer and Link, 2019*; *El-Gebali et al., 2019*). While this TRADD-N domain is the only recognizable globular domain (*Figure 4A*) in the large NCOA6 protein from most animals (*El-Gebali et al., 2019*; *Mahajan and Samuels, 2005*; *Mahajan and Samuels, 2008*), hymenopteran insects additionally possess a C-terminal KH domain (e.g., KYN23013.1, XP_012351037.1; *Figure 4A*, *Figure 4—source data 1*) that might interact with the RNA component of these histone methylase complexes (*Siomi et al., 1993*). Beyond this well-conserved single-copy version, this clade of TRADD domains also shows LSEs of 4–357 proteins in several animals. 25–50% of these proteins possess unique domain architectures (*Figures 3E and 4A–D*). Cnidarians show LSEs of up to four TRADD-N-containing proteins where the domain is typically fused to an N-terminal DED and a variable C-terminal region, which is either a vWA domain or ankyrin repeats related to the substrate-binding region of Fem1 Cullin-E3 ligase complexes (*Figure 4A*; *Dankert et al., 2017*).

In contrast to other molluscs that possess only the NCOA6 ortholog, bivalves (e.g., genus *Crassostrea*) show LSEs of at least 50–66 NCOA6 clade TRADD-N proteins corresponding to 17–20 unique architectural themes (*Figure 4B* and *Figure 4—source data 1*). Comparison of the two related bivalve species with complete genome sequences, *Crassostrea gigas* and *Crassostrea virginica*, shows notable divergence in the architectures of their respective expansions. In *C. virginica*, the primary LSE (~50 proteins) displays a N-terminal region constant region with a cysteine-rich extracellular domain followed by a transmembrane helix, and an intracellular module with a DED and a NCOA6 TRADD-N domain (*Figure 4B*). This is followed by a variable C- terminal region containing one or more of several domains commonly seen in apoptotic/immune proteins (*Figure 4B*): (1) domains of the Death-like superfamily, (2) STAND NTPases, (3) HEPN RNases (*Anantharaman et al., 2013*), and (4) AIG-GTPases, previously studied in plant immunity (*Leipe et al., 2002*), supersecondary-structure-forming repeats (e.g., ankyrins [*Mosavi et al., 2004*] and β-propellers [*Thirup et al., 2013*]; B-box [*Massiah et al., 2007*; *Burroughs et al., 2011*]; and HSP70 domains [*Bracher and Verghese, 2015*]). The remaining set of the *C. virginica* LSE (~10 proteins) consists of proteins with an N-terminal DED-TRADD-N dyad fused to RNA binding KH and RRM domains and multiple macro and C-terminal ART domains resembling PARP-14 that is implicated in antiviral response (*Figure 4B*; *Daugherty et al., 2014*). In contrast, *C. gigas* has only four and two copies respectively of the above architectural themes (*Figure 4B*). Instead, their LSE is dominated by a distinct architecture, with an N-terminal region comprising a HEPN RNase, immunoglobulin repeats, and a DED-TRADD-N combination. C-terminal to these are either WD40 β-propellers (*Chaudhuri et al., 2008*) or in some versions a Mab-21 like cyclic-GMP-AMP synthetase (cGAS) domain (acc: XP_011454060.1), a key component of the cGAS-STING pathway activated by the presence of invasive DNA, followed by TPRs (*Burroughs and Aravind, 2020*; *Figure 4B*).

The cephalochordate *Branchiostoma* is another genus with LSEs of the NCOA6 clade TRADD-N domains; different species show 13–129 proteins (*Figures 3E and 4C*). These are characterized by an N-terminal region with a DED followed by the TRADD-N domain; these in turn are connected by a flexible, low-complexity linker to C-terminal WD40 β-propellers (*Figure 4C* and *Figure 4—source data 1*). The β-propellers vary in number of repeats and show high-sequence variability pointing to their diversification via recombination. The most spectacular expansion of NCOA6 TRADD-N domains (357 proteins) is seen in the sponge *Amphimedon* with up to 50% of the proteins having unique domain architectures (*Figure 4D* and *Figure 4—source data 1*). Thematically, the majority of these architectural types are unified by the presence of an N-terminal unit with 1–3 copies of domains of the Death-like superfamily followed by the TRADD-N domain. This unit is followed by a variable, often fast-evolving, C-terminus with domains belonging to different functional categories (*Figure 4D*). The most common C-terminal domains are ankyrin repeats or a peptide-binding SH3 domain followed by LRRs (*Kay, 2012*; *Kobe and Deisenhofer, 1994*). Alternatively, this region features other signaling and peptide-binding adaptor domains (S/T/Y- protein kinase, MATH, SH2, PDZ; *Tong et al., 1996*; *Zapata et al., 2007*) or potential enzymatic effector domains (caspase, PNPase, the nucleotide phosphodiesterase HD domain; *Aravind and Koonin, 1998b*).

## The DEDD2 and TRADD-like TRADD-N domains

The classic TRADD-N domain clade defined by TRADD is confined to vertebrates (*Figure 3E* and *Figure 4E*). Other than NCOA6 and TRADD, vertebrates typically possess a TRADD-N domain in DEDD2/FLAME-3, a nucleolar protein that interacts with cFLIP, a protein with DED and an inactive caspase domain, to negatively regulate nuclear events during apoptosis (*Zhan et al., 2002*). In both TRADD and DEDD2, the TRADD-N domain is coupled to domains of the Death-like superfamily; in the case of TRADD, it is a C-terminal Death domain, whereas in DEDD2 it is an N-terminal DED (*Zhan et al., 2002*; *Valmiki and Ramos, 2009*). We found a more extensive but sporadic presence of the DEDD2 clade of TRADD-N domains in certain metazoans paralleling the phyletic spread of the NCOA6 clade (*Figure 4F*, *Figure 4—source data 1*; *Anantharaman et al., 2010*). Other than vertebrates, DEDD2 orthologs are also found in echinoderms, molluscs, brachiopods, and arthropods (however, they are lost in insects). Beyond these, LSEs of this DEDD2-clade TRADD-N domains are seen in cnidarians (17–168 copies), *Crassostrea* (14–16 copies), brachiopods (33 copies), echinoderms (18–23 copies), hemichordates (22 copies), and cephalochordates (46–71 copies). Notably, these invertebrate proteins show several domain-architectural parallels to the NCOA6 TRADD-N proteins, and like them, the LSEs of DEDD-2 TRADD-N proteins are also characterized by constant regions combining a Death-like superfamily domain (usually DED) with the TRADD-N domain and variable flanking regions (*Figure 4F* and *Figure 5A*). This clade of TRADD-N domains is also fused to some of the same C-terminal variable domains as those of the NCOA6 clade, which are predicted to function as effectors, albeit in different organisms. For example, in cnidaria, these TRADD-N domains are fused to C-terminal Macro and PARP domains in addition to N-terminal HEPN domains (*Figures 4 and 5*).

However, this clade is characterized by an even greater set of unique domain architectures not observed in the NCOA6 clade (*Figure 4—source data 1* and *Figure 5A*). One architectural pattern, found in several distantly related animals, namely cephalochordates, hemichordates, and cnidarians, features a glycosyltransferase domain of the GT4 family (*Lairson et al., 2008*) preceding the TRADD-N domain and LRRs following it (*Figure 4F* and *Figure 5A*). This clade of TRADD-N domains also shows fusions to the AIG and GBP GTPases, TTC28-like TPR (*Izumiyama et al., 2012*), the oligoadenylate synthase, sacsin, STING, polyubiquitin, and TRIM Ub E3 ligase domains, all which have all been previously observed in other antiviral conflict systems (*Kristiansen et al., 2011*; *Anderson et al., 1999*; *Xue et al., 2018*; *Arimoto et al., 2010Figure 4F*). Similarly, we also observed fusions to known mediators of apoptosis, like the Bcl2 domain, FIIND, and the UPA-Zu5 module (*D'Osualdo et al., 2011*; *Cleary et al., 1986*; *Pathan et al., 2001*; *Tschopp et al., 2003*). Interestingly, other representatives of this clade are fused to a wide range of RNase/RNA-binding domains in cnidarians and brachiopods such as the Piwi module, NYN RNase, HEPN RNase, the interferon-induced SF2 helicase DDX60, the RNAi pathway helicase Mov10, and RLR-like viral RNA receptors (*Anantharaman and Aravind, 2006*; *Burroughs et al., 2014*; *Miyashita et al., 2011*; *Kowalinski et al., 2011*; *Figure 4—source data 1*, *Figures 4F and 5A*).

This pattern of LSEs accompanied by great domain-architectural diversity even between closely related organisms (*Figure 5A*) is largely unprecedented among eukaryotic immune proteins. Hence, we closely examined the genomic neighborhood of these TRADD-N genes and found that in the DEDD2 clade there are several associations with retrotransposons. Moreover, there are certain examples with direct fusions of the TRADD-N-coding genes to retrotransposons and cut and paste DNA transposons related to Mutator/Mule (*Figure 4F*; *Dupeyron et al., 2019*). Hence, we propose that this diversity of architectures with novel fusions is at least in part generated by the transposition of retrotransposons carrying reverse-transcribed segments of conflict-related genes or by the integration of reverse-transcribed transcripts into neighborhoods of TRADD-N coding genes. The resulting gene fusions are then likely to be selected if they provide an advantage against invasive elements. This might explain fusions to whole conflict-related genes that otherwise occur independently of TRADD-N domains in other organisms. Thus, these systems might be seen as paralleling the diversity generated via reverse-transcription in the prokaryotic DGR systems or the immune receptor diversification in *Branchiostoma* (*Wu et al., 2018*; *Huang et al., 2008*).

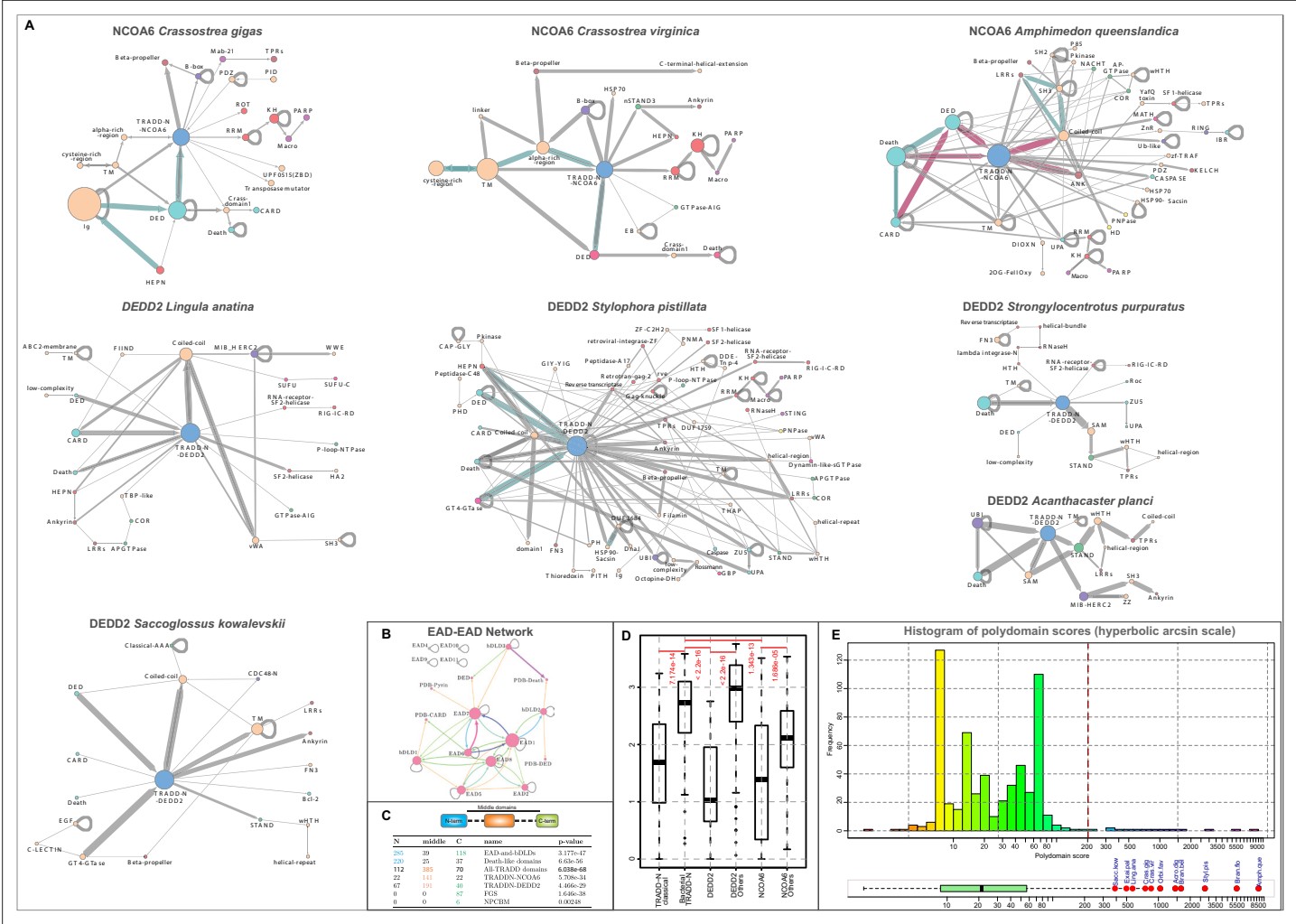

**Figure 5.** Domain architectures and analysis. (**A**) Architectural networks of TRADD-N domains proteins from various metazoan lineages. Nodes represent domains and edges connecting them indicate their adjacency in the same polypeptide with the arrowhead pointing to the C-terminal domain . The thickness of the edge represents the frequency of co-occurrence of the connected nodes in different architectural contexts. (**B**) EAD-EAD search-retrieval network. The network was derived by using the results of various profile-profile (HHpred) searches using the individual nodes (EADs or Death-like domains) as queries. An edge was drawn between two nodes if they were recovered with p-values<0.0001 with the arrowhead pointing to the node recovered in the search. The edge thickness is scaled using the -log$_{10}$(p-value). The network shows that several EADs recover each other and also Death-like domains in searches. (**C**) Chi-squared statistics of the positional bias of the specified domains in their domain architectures. The left three columns show the frequency of domain in the said positions in unique architectures, and the rightmost column gives the probability of this occurring by chance. . (**D**) Boxplot comparing position-specific entropy of sequences from MSAs of various TRADD-N clades. The t-test was used to determine the significance of the difference in mean entropy between the clades under comparison (indicated by the red line above the clades). (**E**) Histogram of the distribution of polydomain scores (PDS) of TRADD-N proteins. Outliers with high PDS, mostly marine organisms, are marked in red. Organism abbreviations are as follows: Sacc. kow.: *Saccoglossus kowalevskii*; Exai. pal.: *Exaiptasia pallida*; Ling. ana.: *Lingula anatina*; Crass. gig.: *Crassostrea gigas*; Crass. vir.: *Crassostrea virginica*; Orbi. fav.: *Orbicella faveolata*; Acro. dig.: *Acropora digitifera*; Bran. bel.: *Branchiostoma belcheri*; Styl. pis.: *Stylophora pistillata*; Bran. flo.: *Branchiostoma floridae*; Amph. que.: *Amphimedon queenslandica*.

## Mechanistic implications of the bacterial Death-like domains and TRADD-N domains

The discovery of unambiguous bacterial versions of the Death-like superfamily and the characterization of the diversity of the TRADD-N domains helps clarify general, unifying mechanisms at the heart of apoptosis and immunity. Although both superfamilies of domains show similar architectural and/or genome contextual linkages to a formidable array of distinct effectors in both prokaryotes and eukaryotes (*Figures 2B and 5A*), they appear to define two distinct mechanistic principles. A key observation in this work is the identification of a common organizational principle unifying the Death-like

domains and the bacterial EADs, thereby placing the former within the spectrum of the radiation of EADs in prokaryotic counter-invader conflict systems. Indeed, in profile-profile searches we were able to recover hits between certain EADs, bDLDs, and members of the metazoan Death-like domains (*Figure 5B*). In bacteria, the common organizational principle is reflected in both the genomic context and domain architectures of the EAD and bDLD proteins. Regarding genomic context, EADs or bDLDs are encoded in multiple copies in the same locus or genome. With respect to domain architectures, they show a statistically significant tendency (p=3.177 × 10$^{-47}$, $\chi^2$-test) to be N- or C-terminally positioned domains that are coupled to other effector and signaling domains (*Figure 5C*).

This terminal positional bias in the domain architectures is also reflected in the metazoan Death-like superfamily (p=6.6 × 10$^{-56}$; *Figures 2B and 5C*, *Supplementary file 3*). Thus, it points to a common role for these domains in bridging different proteins of the apoptotic or immune systems via homotypic interactions to allow the stepwise or aggregated unfurling of the response cascade (*Figure 1C and F*). Alternatively, certain versions of the Death-like superfamily, like the Pyrin domains, self-assemble into large multimeric complexes through a prion-like templating mechanism, resulting in cell death (*Figure 1D*). Comparable filament formation has recently been observed with other effectors (e.g., the TIR domain) and sensors (e.g., the cyclic oligonucleotide-binding STING) in eukaryotic and bacterial immunity, suggesting that it is a more widespread phenomenon (*Li et al., 2012*; *Gentle et al., 2017*; *Morehouse et al., 2020*). Thus, our findings imply that the spectrum of interactions ranging from bridging of effectors and regulators to the self-templated assembly of polymeric complexes is a common function of the EADs and Death-like domains, which has been repeatedly and independently exploited by immune/apoptotic systems of both bacteria and metazoa (*Figure 1C and F*).

The TRADD-N domain shows a rather contrasting positional bias, with a significant tendency to occur between two other globular domains in a single copy within multidomain architectures (p=6 × 10$^{-68}$; *Figure 5C*, *Supplementary file 3*). Further, the newly identified TRADD-N proteins frequently show constant regions combined to regions showing variability in the types of their constituent domains, with the TRADD-N domain usually lying at the junction of the constant and variable parts (*Figure 5A*). As noted above, a Ser residue, which in TRADD-N plays a key role in interacting with its MATH domain partner, is conserved across bacterial and eukaryotic TRADD-N domains (*Figure 3D*; *Park et al., 2000*; *Hsu et al., 1996*). However, the MATH domain is absent in bacteria and there are no corresponding expansions of MATH domains in the organisms with expansions of TRADD-N. Hence, despite the conserved aspect of the interface, the MATH domain is not a universal partner for the TRADD-N domains. Analysis of the Shannon entropy plots of the different families of TRADD-N domains suggests that other than NCOA6 proper, TRADD, and DEDD2 proper, the remaining members of the superfamily have significantly higher mean column-wise entropies. This is an indication that they are under selection for diversification (*Figure 5D*, *Supplementary file 4*). This diversification tracks the variability in the types of domains in the flanking regions. These observations point to a unified model for the action of the TRADD-N domain across these systems that operates via the reconfiguration of protein-protein interactions. Based on the precedence of its role in TRADD and its bridging position in-between flanking domains (*Figure 5C*), we posit that it interacts with both flanking domains to keep them in an inactive state under normal conditions. Upon the reception of an invader- or apoptotic signal, which is either directly sensed by one of the flanking domains or transmitted to it by upstream interactions, the interaction interface of the TRADD-N domain is reconfigured with the flanking domains assuming different conformations. We predict that this switch is mediated by the conserved Ser.

The consequence of these shifting interactions is likely to be twofold: first, it allows recruitment of the TRADD-N protein to a larger complex of apoptotic/immune-related proteins by favoring certain protein-protein interactions (e.g., recruitment of TRADD to the TNFR and in turn the TRAF2 Ub E3 ligase; *Hsu et al., 1996*). This model is also consistent with the repeated, independent association of the TRADD-N domain with the Death-like superfamily domains (more generally EADs) that we observe in both bacteria and eukaryotes. These are likely to recruit the TRADD-N proteins to larger complexes via their homotypic interactions. Second, it is likely to unleash the associated effector domains to attack either self or non-self macromolecules. In some cases, the effector deployment could directly ensue from reconfigured interactions of the TRADD-N domain. In other cases, it might result from a cascading process such as proteolysis. In this regard, it is notable that we found the TRADD-N domain in polypeptides with the FIIND. This predicts that the TRADD-N proteins with ZU5/

FIIND likely undergo autoproteolysis akin to the ZU5 protein PIDD (*D'Osualdo et al., 2011*), which might be critical for effector deployment. Further, the astonishing diversity of effector domains that we find fused to the TRADD-N domains (*Figure 5A*) points to a potentially diversified 'output' that could tackle a wide range of invaders.

## 'Institutionalized' systems and marine ecology-specific immune responses based on the TRADD-N domain

The transcriptional coactivator NCOA6 is the only TRADD-N protein that is retained in a single copy across most of metazoa and also does not contain any conflict-related domains (*Figures 3E and 4A*). Its conservation pattern, including the characteristic Ser residue, suggests that it mediates a switch through protein-protein interactions similar to that proposed for the other TRADD-N domains. However, its above-noted features suggest that, early in metazoan evolution, this TRADD-N domain was coopted and 'institutionalized' for a conserved role in transcription, probably acting as a switch during the recruitment of the histone methylation complex to specific transcription factors. This transcriptional role is paralleled by the prokaryotic versions associated with ECF-sigma factor HTH domain (*Heimann, 2002*).

The apoptosis/immune-related TRADD-N domains show a contrasting phyletic pattern between vertebrates and invertebrates. The former show low numbers of TRADD-N domains (*Kysela et al., 2016*; *Dunin-Horkawicz et al., 2014*; *Rokas, 2008*; *Ameisen, 2002*), and those involved in apoptosis/immunity are nearly always the orthologs of DEDD2 and TRADD. In contrast, phylogenetically distant lineages of invertebrates show expanded repertoires of diverse TRADD-N domains, which are typified by striking lineage-specific differences in domain architectures, even between closely related species. Further, such expansions are restricted to marine species, namely cephalochordates, hemichordates, echinoderms, molluscs, brachiopods, cnidarians, and sponges (*Figures 3E and 5E*). Beyond this, it is notable that every marine group showing such LSEs of TRADD-N domains are either largely sessile as adults (e.g., sponges, cnidarians, brachiopods, bivalve molluscs, and hemichordates) or show slow-moving massive herding behavior (e.g., sea cucumbers and starfishes) or dense seasonal aggregation (amphioxus) in marine environments (*Könnecker and Keegan, 1973*; *Marquet et al., 2018*; *Craeymeersch and Jansen, 2019*; *Kenchington and Hammond, 2009*; *Frankenberg, 1968*; *Piper, 2015*). This is exemplified by the fact that arthropods or cephalopod molluscs, which, like vertebrates, tend to be highly motile, have very few (e.g., DEDD2) or no TRADD-N domains. We quantified this tendency using the previously devised metric the polydomain score (PDS) (*Schäffer et al., 2020*) that captures both the prevalence (tendency to occur as LSEs) and the domain architectural complexity of a class of proteins in a given organism as a single number (*Figure 5E*; see Materials and methods). Whereas the majority of organisms show TRADD-N proteins with comparable PDS, the above-mentioned marine metazoans are dramatic outliers, with significantly higher PDS than the mean of 74 (often orders of magnitude greater; range of 300–9000; $p < 10^{-6}$; *Figure 5E*; *Supplementary file 5*). Together, these observations suggest that the TRADD-N proteins are a feature of a peculiar form of immune response that is strongly selected in certain marine environments, like reefs or shallow ocean floors, where these animals aggregate (*Palmer and Traylor-Knowles, 2012*).

The diversification of the TRADD-N-associated variable effector domains, between even closely related species (e.g. *Figure 5A*, first two networks), is notable. More specifically, we observed that they feature (1) known RNA receptors (e.g., RLR), RNA-binding domains (e.g., KH, RRM), and RNA-targeting endonucleases (e.g., HEPN, NYN). These suggest RNA viruses or retroviruses/mobile retrotransposons to be potential stimulating signals as well as targets for these proteins (*Figures 4 and 5A*). (2) Superstructure-forming repeats (e.g., β-propellers and ankyrins) associated with the TRADD-N domains suggest that they might directly recognize invader molecules to activate the TRADD-N conformational switch to unleash associated effectors (*Figures 4 and 5A*). (3) TRADD-N proteins in receptor-like architectures in the bivalve *Crassostrea*. This suggests that external stimuli in the form of direct adherence of pathogens or activated immunocytes might also trigger the TRADD-N switch. In this case, the pathogens could also include cellular forms, including infectious tumor cells that have been reported in these molluscs (*Collins and Mulcahy, 2003*; *Metzger et al., 2015*; *Oprandy et al., 1981*). (4) Ub-system domains like Ub and Ub-ligases (*Burroughs et al., 2012a*; *Burroughs et al., 2012b*): these suggest responses involving ubiquitination of invader proteins as well as recognition of invader proteins ubiquitinated by other E3 ligases involved in anti-invader defense. This array of

effector domains is consistent with both direct targeting of invader molecules and containment of the invader via apoptosis. Moreover, their variability also indicates an arms-race scenario potentially driven by rapidly evolving resistance to the effectors in the invaders either from sequence divergence in the targets or due to the effectors being nullified by invader-derived inhibitors.

Finally, it is notable that the TRADD-N domains are shared not just between disparate metazoans with similar ecology but also disparate multicellular bacteria with similar sessile aggregations or colonies (*Figures 3 and 4*). Hence, they are likely to mediate a mechanism of immunity that was likely acquired by animals from these multicellular bacteria and fixed due to similar selective pressures. Given the above observations, we posit that marine aggregations might be susceptible to a variety of pathogens that might rapidly spread through them. In these animals, the TRADD-N proteins were probably selected for as a confluence for a diversified repertoire of immune mechanisms of disparate provenance that function independently in other metazoans. Intriguingly, these phylogenetically disparate metazoans also uniquely share certain domain architectures of TRADD-N proteins to the exclusion of the rest. For example, the glycosyltransferase GT4-TRADD-N combination (see below) is shared by cnidarians, cephalochordates, and hemichordates and no other metazoan species (*Figures 4 and 5A*). This raises the possibility of some of these immune genes spreading through lateral transfer in marine environments. This possibility is strengthened by the above-noted association with retrotransposons, some which have been noted to jump between different marine species (*Metzger et al., 2018*).

## TRADD-N proteins help identify novel effector-deployment systems

Spurred by the observation that the metazoan TRADD-N proteins bring together a wide array of potential effector domains, we carefully analyzed these to detect previously unrecognized effector-deployment systems. We present below two interesting examples that were identified by this analysis.

### Glycosyltransferase-4 (GT4) domains

The first of these is the glycosyltransferase GT4 domain, with two Rossmann fold subdomains (*Martinez-Fleites et al., 2006*), that shows numerous associations with the TRADD-N domain across diverse invertebrates (*Figure 4F*). GT4 domains are also found independently of the TRADD-N domain in a massive LSE of over 500 paralogs in the coral *Acropora digitifera*. Beyond the GT4 domain, these proteins display high variability of domain architecture with fusions to several other domains suggestive of roles in biological conflicts (*Figure 6A*, *Figure 6—source data 1*). We found that the metazoan GT4 domains are closest to the versions found in a range of bacterial conflict-related contexts (*Figure 6B*) such as the above-noted GreA/B-C systems. Homologous GT4 domains are also found in the effector position in the recently described MoxR-vWA-dependent VMAP and caspase-dependent CATRA ternary conflict systems (*Kaur et al., 2020*; *Burroughs and Aravind, 2020*). Other than these associations, GT4 domains occur in specialized giant actinobacterial secreted toxins related to the effectors from polymorphic toxin systems deployed in inter-bacterial conflicts (*Figure 2—source data 1*, *Figure 6B*, bottom row, *Figure 6—source data 1*; *Zhang et al., 2012*). Here, the GT4 domains are combined with an array of HTH domains, other toxin domains such as ARTs and metallopeptidases, and SecA-like ATPase and major facilitator superfamily (MFS) transporter domains that are predicted to facilitate their secretion (*Sharma et al., 2003*; *Marger and Saier, 1993*).

Our analysis revealed several further bacterial GT4 domains that are predicted effectors in counter-invader conflict systems. Several of these are found frequently fused to various classical STAND NTPase modules, such as AP-ATPase, NACHT, and SWACOS (*Leipe et al., 2004*), often in a C-terminal position comparable to the above-reported fusions of related STAND NTPases to different lectin fold domains (*Figure 2H,I*). A subset of these STAND NTPase-GT4 proteins also contain additional N-terminal effector domains, namely TIR and caspase (*Figure 6B*, last column). Beyond these, the GT4-STAND NTPase domain fusions sometimes occur as part of proteins encoded by a distinct, mobile, bacterial two-gene operon (*Figure 6C*). The constant core of both proteins encoded by this operon are STAND NTPase modules. These STAND NTPases define a novel family that is closer to the recently reported CR-ATPases (*Zhang et al., 2016*), the mitochondrial apoptotic ribosomal protein S29/Dap3 (*Kim et al., 2007*), and another mitochondria-associated STAND NTPase RNA12 (*Hanekamp and Thorsness, 1996*) than to the above-mentioned classic STAND domains (*Zhang et al., 2016*). While both proteins possess a conserved arginine finger, suggesting that they form a toroidal hetero-multimer, the NTP-binding motifs of one of these STAND proteins have eroded, indicating that it is inactive like

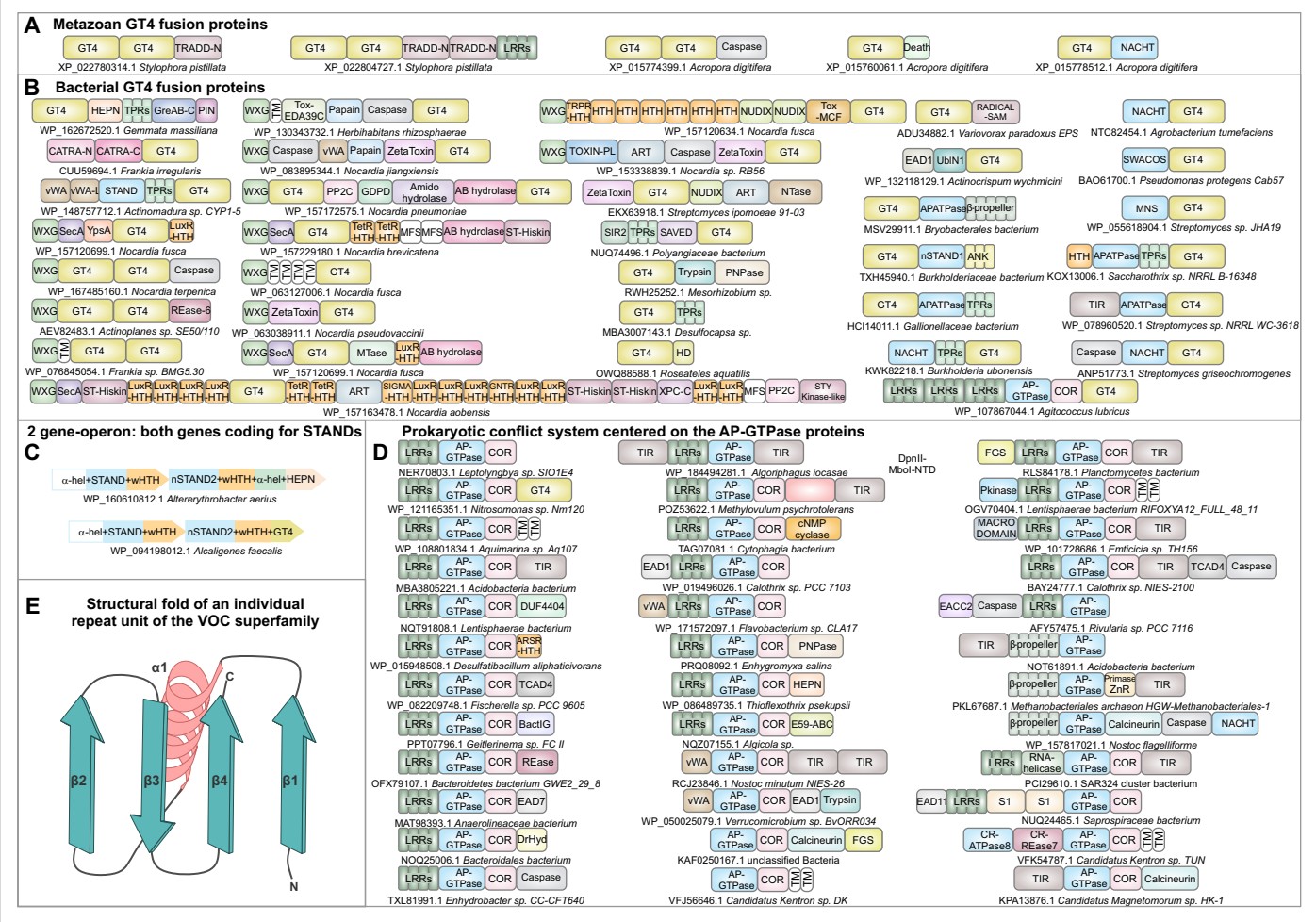

**Figure 6.** Domain architectural associations of the GT4 glycosyltransferase domain in (**A**) representative metazoans and (**B**) prokaryotes. (**C**) Predicted counter-invader two-gene systemwith each gene encoding a novel STAND NTPase (nSTAND2). (**D**) Prokaryotic counter-invader systems centered on the AP-GTPase proteins. (**E**) Topology diagram of the structural fold of an individual repeat unit of the vicinal oxygen chelate superfamily illustrating its core secondary structure units.

The online version of this article includes the following source data for figure 6:

**Source data 1.** Gene neighborhoods and domain architectures of the GT4 glycosyltransferases, the nSTAND2 and prokaryotic AP-GTPases.

the iSTAND and iSTAND2 versions (*Figure 2—source data 2*). This inactive STAND module is fused to variable, predicted effector modules that are most commonly either a GT4 domain or a HEPN RNase domain (*Figure 6C*).

The GT4 domain is also found in the effector position of another prokaryotic conflict system centered on the AP-GTPase proteins (*Figure 6D*, *Figure 6—source data 1*), which are homologs of their metazoan counterpart, the apoptotic DAP protein kinase (*Kawai et al., 1999*; *Bialik and Kimchi, 2012*). In these proteins, the AP-GTPase and associated COR domain are fused to rapidly evolving N-terminal leucine-rich repeats akin to those found in the intracellular pathogen-recognition module of the NACHT clade in animals and the AP-ATPase clade in plants (*Figure 6D*; *McHale et al., 2006*; *Correa et al., 2012*). Their C-terminal region shows an assortment of variable domains, which is a kinase in the case of the metazoan DAP. In bacteria, this position is occupied by at least 24 different domains (*Figure 6D*), which, in addition to the GT4 domain, include the usual array of frequently found effectors from conflict systems such as the TIR, DRHyd, caspase, trypsin and FGS domains, or different EADs. A single bacterium can possess multiple copies of these proteins, each with a different effector domain (e.g., *Haliscomenobacter hydrossis*, a filamentous bacteroidetes possess 11

paralogous copies), suggesting a broad-based strategy to counter immune evasion and inhibition of effectors by the invaders.

Together, these observations suggest that the GT4 domain is a potential effector that is used in a variety of bacterial conflict systems both in intracellular counter-invader and inter-organismal conflict. Via lateral gene transfer, it has also been 'drafted into' and expanded in the immune systems of certain metazoans. Biochemically characterized metabolic versions of the GT4 family modify a variety of substrates such as glycoproteins, lipids and sugars, and metabolites like mycothiol (*Campbell et al., 1998*; *Vetting et al., 2008*). This, along with the expansion in animals, indicates that they might act in anti-pathogen conflicts via modification of the invader molecules to either directly neutralize their interactions or render them available to binding by lectin-like domains of conflict systems (see above). Alternatively, they could block interactions between invader and host molecules by glycosylation of residues at the interface.

## FAM124-type VOC superfamily domains

We found the approximately 500-amino-acid-long, so-called FAM124 domain coupled to the TRADD-N domain in metazoans (*El-Gebali et al., 2019*; *Figure 3F*). Several of these contain a conserved extension with a predicted caspase site, indicating that they are possibly regulated as part of a proteolytic cascade by cleavage at this site. The prototypic member of this family, FAM124B, which only has the FAM124 domain, is a nuclear protein that interacts with SWI2/SNF2 ATPase-driven chromatin-remodeling complexes such as CHD7 and CHD8 (*Batsukh et al., 2012*). Secondary structure prediction based on a multiple alignment indicated that the FAM124 domain is made of up of two subdomains with comparable secondary structure (Figure—source data 2, *Figure 6E*). Of these, the C-terminal β-α-β-β-β subdomain retrieves vicinal oxygen chelate (VOC) superfamily dioxygenase/glyoxylase domains in iterative sequence-profile searches (e = $2 \times 10^{-4}$, iteration 3). Members of this superfamily have two homologous tandem β-α-β-β-β structural units, with the β-sheets of two units (derived from the same or different polypeptides; *Figure 6E*, *Figure 2—source data 2*) dimerizing to form an incomplete barrel-like structure (*He and Moran, 2011*; *Gerlt and Babbitt, 2001*). Given that FAM124 shows a N-terminal region with a comparable secondary structure to the C-terminal unit that can be unified with the VOC superfamily, as well as the conserved motifs that interact with the linker between the two tandem subdomains, we propose that the N-terminal region represents the first repeat (*Figure 6E*).

In most VOC superfamily members, the tandem units cooperate to coordinate divalent metal ions through residues in β1 and β4 of the two units and two vicinal oxygen atoms in the substrate. While these residues differ between VOC families, their positions are conserved across the families (*He and Moran, 2011*; *Gerlt and Babbitt, 2001*; *Armstrong, 2000*). Many FAM124 domains possess a highly conserved histidine and glutamate in the regions corresponding to β1 of the N- and C-terminal structural units, and glutamines in the β4 of the N- and C-terminal structural units (*Figure 2—source data 2*). Hence, the versions with these residues have the potential to coordinate a metal ion and exhibit catalytic activity. The VOC superfamily catalyzes diverse enzymatic reactions such as isomerizations, epimerizations, and oxidative ring cleavage (extradiol dioxygenase) (*Gerlt and Babbitt, 2001*). Due to this diversity, the precise activity of FAM124 remains unclear. Based on its association with the CHD7 and CHD8 complexes, it is tempting to suggest that its enzymatic activity might be directed towards chromatin proteins, such as in the isomerization or restoration of glyoxalated amino acid side chains. Those TRADD-N-fused FAM124 domains, which lack the above-noted conserved residues (*Figure 6E*), could still bind target proteins like the active versions.

## Discussion

### The newly identified conflict systems provide insights regarding the evolution of immune and apoptotic systems

In prokaryotes, highly regulated conflict systems, such as those described here and previously, are strongly correlated with a multicellular habit across phylogenetically distant lineages (*Kaur et al., 2020*). The thematically similar but lineage-specific domain architectures in both bacteria and multicellular eukaryotes suggest that a basic 'vocabulary' of apoptosis-related domains (*Dunin-Horkawicz et al., 2014*; *Aravind et al., 2001*; *Kaur et al., 2020*; *Hofmann, 2019*) was recombined with each

other and with other immune-related domains to constitute domain architectures with similar syntax in each of the disparate lineages that possess them. The presence of similarly expanded complements of TRADD-N and Death-superfamily proteins across distantly related aggregating marine invertebrates is one such case (*Figures 4 and 5*). This argues for both the existence of a grammar of functional interactions between these domains and strong selective advantages for coupling immunity and apoptosis, especially in multicellular organisms across the tree of life. More specifically, our observations also elucidate the emergence of specific types of components in the tangled history of immune and apoptotic systems. We consider these in greater detail below.

## NTPase switches, sensors and adaptors

More than two decades ago, it was observed that the STAND NTPases are a common denominator of the apoptotic systems in animals, plants, and fungi, which are shared with bacteria (*Aravind et al., 1999*). The vast radiation of these NTPases in bacteria, along with the nesting of the eukaryotic versions within these bacterial radiations, indicated that they have been repeatedly acquired by eukaryotes (*Aravind et al., 2001*; *Koonin and Aravind, 2002*). Some examples like the NACHT telomerase subunit TLP1 and the mitochondrial ribosomal protein S29/Dap3 were early acquisitions (*Koonin and Aravind, 2000*; *Kissil et al., 1999*), whereas other acquisitions are limited to certain lineages (*Koonin and Aravind, 2002*). Over this period, structural studies revealed how the metazoan AP-ATPases and NACHTs form toroidal complexes, such as the apoptosome and the inflammosome, that form a platform for NTP-regulated effector deployment in related processes in apoptosis and immunity (*Riedl and Salvesen, 2007*; *Duncan et al., 2007*; *Latz, 2010*; *Cain et al., 2000*). In another direction, these studies also established that STAND NTPases belong to the CDC6/Orc clade of AAA+ NTPases, whose archetypal members mediate the assembly of the replication origin complexes in the archaeo-eukaryotic lineage (*Leipe et al., 2004*). Further, the recent study of the CR-ATPases, which are ATPase domains found in secreted effectors deployed in eukaryotic inter-organismal conflict, revealed the evolutionary trajectory of the STAND NTPases from CDC6/Orc-related ATPase domains of the transposases of certain mobile DNA elements and Mu-like bacteriophages (*Zhang et al., 2016*). Similarly, it has also become clear that other NTPase regulatory domains, namely AP-GTPases and AIG-GTPases, also have an ultimately bacterial origin (*Aravind et al., 2001*; *Leipe et al., 2002*).

In bacteria, STAND NTPases are widely distributed and are implicated in several functions other than apoptosis proper, for example, as sensory switches in transcription regulation and antiviral defense (*Leipe et al., 2004*). Recently, we uncovered a class of highly regulated prokaryotic conflict systems, the ternary systems, which are enriched in multicellular bacteria (*Kaur et al., 2020*). These systems, together with those reported in the current work, firmly establish the connection of a subset of bacterial STAND proteins to immunity-linked apoptotic responses. A notable point emerging from these studies is the deployment of both active and inactive versions that span the evolutionary range of the STAND clade of NTPases from the more basal clades closer to the transposase ATPases and CR-ATPases to the well-known classical clades such as AP-ATPase, NACHT, and SWACOS (*Leipe et al., 2004*; *Kaur et al., 2020*; *Zhang et al., 2016*). The active versions show all the hallmarks of forming apoptosome/inflammosome-like toroidal complexes for effector deployment, akin to their eukaryotic counterparts.

In contrast, the inactive versions do not have functional precedents in the well-studied eukaryotic STAND systems. However, in the related hexameric ORC complex of eukaryotes, only two of the ATPase domains are active, though both the active and inactive versions bind DNA (*Foss et al., 1993*). Other STAND NTPases like the ribosomal S29/Dap3 GTPase have been shown to bind single-stranded RNA as monomers (*Kissil et al., 1999*; *Waltz et al., 2019*; *Berger et al., 2000*), an activity also probably possessed by the eukaryotic RNA12 STAND protein. We found at least three distinct versions of inactive STAND domains, two in the ternary systems (iSTAND and iSTAND2) and one in the system with GT4 and HEPN effectors (*Kaur et al., 2020*). In the case of iSTAND and iSTAND2, they are predicted to form the invader-sensing component. Hence, we propose that more generally the inactive STAND domains retain the ancestral nucleic-acid-binding role to act as nucleic acid receptors. Thus, they are predicted to function as the bacterial counterparts of metazoan nucleic acid receptors such as the RLRs and the superfamily-2 RNA helicase receptors (e.g., IFIH1) (*Chattopadhyay and Sen, 2017*; *Kowalinski et al., 2011*; *Yu et al., 2018*). Therefore, the STAND NTPases appear to have taken two trajectories – first, as NTP-dependent switches for regulating the assembly of multimeric

apoptotic/immune complexes and, second, as nucleic acid sensors, echoing the nucleic acid-binding capabilities of their ancient ORC/CDC6-like predecessors. In contrast, the AP-GTPases appear to have remained only as switches in both bacteria and metazoa.

This thematic equivalence among the invader-sensing modules of prokaryotic and eukaryotic conflict systems is also reinforced by our discovery of distinct multiple lectin fold sensors, viz., NPCBM (*Rigden, 2005*) and FGS (*Alayyoubi et al., 2013*), in prokaryotic conflict systems (*Figure 2H,I*). These parallel the use of the SPRY domain with the concanavalin lectin fold in the widespread eukaryotic TRIM conflict systems (*D'Cruz et al., 2013*; *Perez-Caballero et al., 2005*). While we have not yet observed any prokaryotic conflict systems deploying SPRY-like domains, it is notable that, like the FGS domain which was first reported in variable tail proteins of phages such as BPP1 (*Liu et al., 2002*), the SPRY domain too has its origin in bacteriophage tail proteins (*Iyer et al., 2021*; *Mackrill, 2012*; *Figure 2—source data 2*).

## Bacterial provenance of apoptotic adaptor domains

As with the regulatory NTPases, studies over the past two decades have shown that much of the core 'domain vocabulary' of eukaryotic apoptotic systems, like TIR, PNPases, CIDE (CAD/DFF40)-like HNH endonucleases, caspases, ZU5 autopeptidases (e.g., in PIDD1) and components of the metazoan ASK signalosome, have their ultimate origins in bacterial conflict systems (*Zhang et al., 2012*; *Aravind et al., 1999*; *Koonin and Aravind, 2002*; *Burroughs et al., 2015*; *Burroughs and Aravind, 2020*; *Zhang et al., 2011*; *Aravind et al., 2000*). This work and the earlier-reported ternary systems (*Kaur et al., 2020*) again show that these domains are likely to be used comparably to the eukaryotic versions in bacterial counter-invader systems, especially in multicellular bacteria. However, the provenance of the adaptor domains such as those of the Death-like superfamily and TRADD-N, which are so critical to metazoan apoptosis (*Park et al., 2007*; *Park et al., 2000*), remained mysterious. Here, we show that the eukaryotic Death-like and TRADD-N domains are found in bacteria. Notably, they possess domain architectures comparable to their counterparts in metazoan apoptotic/immune systems (*Figures 3–5*). Moreover, the Death-like domains are part of a larger radiation of bacterial EADs that are adaptor components of bacterial counter-invader systems (*Kaur et al., 2020*). This establishes that not just the enzymatic effectors but also key non-enzymatic adaptors were already in place in the bacterial systems that couple immunity and apoptosis.

## Conclusions

The findings presented in this work offer a more complete picture of the intertwined evolutionary connections between immune and apoptotic systems of eukaryotes and prokaryotes. Phylogenetically distant multicellular or social forms are threatened by intracellular invaders/pathogens that can rapidly spread from cell to cell across the ensemble. This, coupled with inclusive fitness due to their clonal nature or close relatedness, favors the origin of apoptotic mechanisms of defense that limit the invasions to the initially infected cells. Other components of the metazoan toolkit, such as components of the calcium stores system and adhesion molecules, have also been reported to have their roots in bacteria with multicellular tendencies (e.g., cyanobacteria) (*Schäffer et al., 2020*; *Aravind et al., 2003*). Hence, special habitats that bring together colonial eukaryotes and bacteria (e.g., stromatolites) might have been the epicenters of the origin and spread of multicellularity (*Lyons and Kolter, 2015*; *Schirrmeister et al., 2011*; *Bosak et al., 2013*).

The tracing of the classical metazoan Death-like domain to the bacteria and the identification of the TRADD-N diversification adds to the functional understanding of these adaptor domains. Specifically, the predictions made regarding the mode of action of the TRADD-N domain could help understand the poorly understood immune systems of invertebrates. The prediction of shared immune mechanism across diverse invertebrates with comparable ecological features could help understand epidemics in marine ecosystems. Finally, we also believe that the predicted nucleic acid sensor and catalytic domains such as GT4 described here might help develop unique biochemical reagents to detect and modify biomolecules.

## Materials and methods

### Comparative genomics

The NCBI Taxonomy database was used to obtain the names and ranks of taxa. For contextual analysis of prokaryotic gene neighborhoods, the GenBank genome files corresponding to unique GenBank genome assemblies (GCA ids) were used as the starting material. Specific neighborhoods were extracted using a Perl script that reports upstream and downstream genes of the anchor gene. Proteins encoded by these genes were then clustered using BLASTCLUST (https://ftp.ncbi.nih.gov/ blast/documents/blastclust.html) (RRID:SCR_016641) to identify conserved gene neighborhoods based on conservation between different taxa. Additional filters were then used to output valid neighborhoods for further analysis: (1) nucleotide distance constraints (generally 50 nucleotides), (2) conservation of gene directionality within the neighborhood, and (3) presence in more than one phylum.

### Sequence-based analysis

PSI-BLAST (RRID:SCR_001010) (*Altschul et al., 1997*) and JACKHMMER programs (RRID:SCR_005305) (*Eddy, 2009*) were used to carry out iterative sequence profile searches. The BLASTCLUST program was used for clustering for classification or filtering of nearly identical sequences. The program takes both the length of the pairwise alignment (L) and measure of similarity, that is, bit-score (S), as inputs; these were changed as per the degree of required clustering. As an example, the length (L) and bit-score (S) parameters for clustering near identical proteins were set at L = 0.9 and S = 1.89.

HMMsearch with an HMM constructed from an alignment or iteratively built using JACKHMMER from single sequences were used as an alternative search strategy. The searches were run against either (1) the non-redundant (nr) protein database of the National Center for Biotechnology Information (NCBI) frozen at October 1, 2020; or (2) the same database clustered down to 90, 70, or 50% similarity using the MMseqs program (RRID:SCR_008184) (*Hauser et al., 2016*); or (3) a custom database of 7423 complete genomes extracted from the NCBI RefSeq database.

Profile-profile searches were run using HHpred (RRID:SCR_010276) (*Zimmermann et al., 2018*) against (1) HMMs derived from PDB, (2) Pfam A models (*Finn et al., 2016*), and (3) a custom database of alignments of diverse domains curated by the Aravind group. The boundaries of several domains in Pfam were corrected and expanded with additional divergent members that were not found by the original Pfam models to improve detections. Given the rapid sequence divergence of proteins in biological conflict systems, improved detection of homology both in terms of range (i.e., detection of more homologs) and depth (i.e., detection of more distant homologs) is an important issue. The former can be problematic because of the continuum of sequence divergence between homologs; hence, a single starting point might not be sufficient to detect all homologs in iterative sequence searchers. The latter arises from absence of 'bridging' sequences in the current database between two distant homolog groups. Hence, to achieve 'sensitivity' on both these accounts we used curated, successively constructed multiple alignments that encompass an increasing diversity of sequences to construct the profile for RPS-BLAST, HMM, and profile-profile HHpred searches. All new alignments that were used or generated in this study are provided in *Figure 2—source data 2*. Multiple sequence alignments were built using the Kalign (RRID:SCR_011810) (*Lassmann et al., 2009*), HMM3 or Muscle (*Edgar, 2004*) programs, and manually improved based on profile-profile and structural alignments. Secondary structure prediction was done using the JPred program (RRID:SCR_016504) (*Cole et al., 2008*).

As significant patterns in sequence divergence help identify proteins in conflict systems, we used statistically significant differences in mean column-wise Shannon entropy of alignments as a measure tested using the t-test (*Krishnan et al., 2018*; *Burroughs et al., 2017*). Position-wise Shannon entropy (H) for a given column of the multiple sequence alignment was calculated using the equation

$$H = - \sum_{i=1}^{M} P_i \log_2 P_i$$

where P is the fraction of residues of amino acid type i, and M is the number of amino acid types.

Phylogenetic trees were constructed using the FastTree (*Price et al., 2010*) (RRID:SCR_015501) and the IQ-TREE (*Nguyen et al., 2015*) (RRID:SCR_017254) programs. For the former, the JTT model was used with 10 rate categories of sites and the -slow option for an exhaustive tree search. For the

latter, an exhaustive model search determined the WAG model with four gamma-distributed rate categories and empirical state frequency determined from the alignment as the most appropriate for tree construction.

## Other data analysis

Structure similarity searches were conducted with the DaliLite program (RRID:SCR_003047) (*Holm, 2019*) run against the PDB database clustered at 75% sequence identity. Structure-based trees were constructed by converting Z-scores from an all-vs-all search of the compared structures into a distance matrix and performing average linkage clustering. Structures were visualized rendered with the PyMol (http://www.pymol.org) (RRID:SCR_000305) and MOL* programs (http://molstar.org). All other data processing (knitr and dplyr libraries), network analysis (igraph libraries), and graph visualization were done in the R language. Delimited datasets used in these analyses are available in the source data files and at ftp://ftp.ncbi.nih.gov/pub/aravind/Death_TRADDN/. PDS were computed thus (*Schäffer et al., 2020*): organism PDS were calculated as follows: if $c\,(o,p)$ counts the number of proteins of a given protein domain $p$ in an organism $o$, $P$ is the set of all proteins studied, and $O$ is the set of all organisms studied, then the PDS for an organism $o \in O$ is defined as

$$PD\,(o) = \sum_{p \in P} c\,(o,p) \cdot \left(f\,(p) - \bar{f}\right)$$

where $\bar{f}$ is the mean of $f\,(p)$ for all proteins $p \in P$, and $f\,(p)$ is defined as

$$f\,(p) = \log_2 \left( \frac{\sum_{o \in O} c\,(o,p)}{\sum_{q \in P} \sum_{o \in O} c\,(o,q)} \right)$$

The hypergeometric test was used to test for association of systems described herein with multi-cellularity in prokaryotes thus (*Kaur et al., 2020*): each prokaryotic organism in the above-mentioned complete genome database assembled from NCBI GenBank/RefSeq was systematically assessed and assigned a multicellularity flag (True, False, NA: when not known) using all available information obtained from the Bergey's Manual of Systematic Bacteriology (*Whitman, 2015*) and the available publications on the individual taxon. This allowed scoring of 7538 organisms in the database (*Supplementary file 6*) accounting for almost all prokaryotes except the candidate phyla radiation for which there is no information (*Lyons and Kolter, 2015*). This gave us the background frequency of organisms with or without a multicellular habit, which we used to test significance of the associations of the systems reported here for multicellularity using the hypergeometric distribution implemented in the phyper function of the R language. For this test, the four input values were: q = the number of organisms containing a copy of a given system that score as multicellular; m = the total number of multicellular organisms in the database; n = the total number of non-multicellular organisms in the database; and k = the total number of organisms with the given system drawn without replacement from the total set in the database. The $\chi^2$ test for the bias in location of the domains in architectures was performed using its implementation in the R language in form of the chisq.test command.

## Acknowledgements

This research was supported by the Intramural Research Program of the NIH, National Library of Medicine.

## Additional information

### Funding

| Funder | Grant reference number | Author |
|---|---|---|
| National Library of Medicine | LM000084-20 | Gurmeet Kaur |

| Funder | Grant reference number | Author |
|---|---|---|

The funders had no role in study design, data collection and interpretation, or the decision to submit the work for publication.

## Author contributions

Gurmeet Kaur, Data curation, Formal analysis, Validation, Visualization, Writing – original draft; Lakshminarayan M Iyer, Conceptualization, Data curation, Formal analysis, Software, Validation, Visualization, Writing – original draft; A Maxwell Burroughs, Data curation, Formal analysis, Investigation, Validation, Writing – review and editing; L Aravind, Conceptualization, Formal analysis, Funding acquisition, Investigation, Methodology, Software, Supervision, Validation, Visualization, Writing – review and editing

## Author ORCIDs

Gurmeet Kaur ⓘ http://orcid.org/0000-0003-2442-1515
Lakshminarayan M Iyer ⓘ http://orcid.org/0000-0002-4844-2022
A Maxwell Burroughs ⓘ http://orcid.org/0000-0002-2229-8771
L Aravind ⓘ http://orcid.org/0000-0003-0771-253X

## Decision letter and Author response

Decision letter https://doi.org/10.7554/eLife.70394.sa1
Author response https://doi.org/10.7554/eLife.70394.sa2

# Additional files

## Supplementary files

• Supplementary file 1. p-Values of profile-profile search (HHPRED) hits recovered with various effector-associated domains and bacterial Death-like domains as queries.

• Supplementary file 2. Hypergeometric distribution test for association of various conflict systems described in the text with multicellular prokaryotes.

• Supplementary file 3. Chi-squared statistics of the positional bias of major domains described in the study. The left three columns show the frequency of domain occurrence in the various positions in unique architectures, and the probability of this occurring by chance is given in the rightmost column.

• Supplementary file 4. Position-specific entropy of sequences from multiple sequence alignments of various TRADD-N clades.

• Supplementary file 5. Polydomain scores of various TRADD-N-domain-containing species.

• Supplementary file 6. Multicellular status of all organisms included in analysis of the significance of systems overrepresentation in multicellular prokaryotic genomes. The multicellular flag is defined as either True, False, or NA (not enough information available).

• Transparent reporting form

## Data availability

All data generated or analysed during this study are included in the manuscript and supporting files. Source data files have been provided for Figures 2, 3, 4, 5, 6.

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
