## [Decision Letter]

[Editors' note: this paper was reviewed by Review Commons.]

**Acceptance summary:**

This paper presents a complex protein and genome sequence-analytic study that connects apoptotic biomolecular systems and counter-invader immunity systems. The authors identify true bacterial homologs of the Death-like 8 and TRADD-N superfamilies and show that these domains have emerged as part of a radiation of effector-associated α-helical adaptor domains. They also discover new counter-invader systems using intracellular invader-sensing lectin-like domains (NPCBM and FGS), glycosyltransferase-4 family sugarmodification domains, inactive NTPases serving as nucleic-acid-receptors and invader-sensing GTPase switches.

---

## [Author Response]

Reviewer #1The authors present a very complex protein and genome sequence-analytic study that connects apoptotic biomolecular systems and counter-invader immunity systems: (1) They identify true bacterial homologs of the Death-like 8 and TRADD-N superfamilies. (2) These domains have emerged as part of a radiation of effector-associated α-helical adaptor domains that likely, via homotypic interactions, bring together diverse effector and signaling domains.(3) The TRADD-N domain links effector deployment to invader-sensing in multicellular bacterial and metazoan counter-invader systems. (4) They discover new counter-invader systems using intracellular invader-sensing lectin-like domains (NPCBM and FGS), glycosyltransferase-4 family sugarmodification domains, inactive NTPases serving as nucleic-acid-receptors and invader-sensing GTPase switches.Significance:The discoveries described in this manuscript are fundamental for the understanding or evolution and the network of apoptotoc/immunity biomolecular systems. This is a very good work.Place in the literature:This will be a seminal paper.Audience:All life scientists with general interest.My expertise:Bioinformatics, sequence analysis, gene function discovery, natural products.

We thank the reviewer for the positive comments. We address the specific comments and our responses to them below.

1. Several times, the authors mention "sensitive sequence analysis" without actually telling what they have in mind. Clearly, the approach involves including hits even if sequence statistical measures are not fully supportive/borderline. This is not an error if other indicators (hydrophobic patterns, gene context in procaryotes, important functional residues, etc.) support the conclusion. It would be great if the authors transparently showed when their sequence analysis needed to become extra-"sensitive".

By sensitivity of sequence analysis we mean the capacity for improved detection of homology both in terms of range (i.e. detection of more homologs) and depth (i.e. detection of more distant homologs). The former is an issue because of the continuum of sequence divergence between homologs; hence, a single starting point might not be sufficient to detect all homologs in iterative sequence searchers. The latter arises from absence of “bridging” sequences in the current database between two distant homolog groups. Both are specifically important when studying proteins in biological conflict systems due to their rapid sequence divergence. Sensitivity can be increased by using curated, successively constructed multiple alignments that encompass an increasing diversity of sequences to construct the profile for RPS-BLAST, HMM and profile-profile HHPRED searches. We have now added more details to the Materials and methods explaining this (page 26, lines 7-16).

2. As a reader, I would appreciate a compression package of all the alignments, domain definitions, etc. in a computer-readable form (e.g., in aln format) as supplementary material. This would greatly help to make the work reproducible.

We now provide in the supplementary material a compressed archive containing all alignments generated in this study as text alns. We will also be also submitting these alignments to the Pfam database for wide dissemination across the community upon publication.

Minor Comments:The manuscript is written clearly, all conclusions are backed by data. Yet, it would be helpful to have Results and Discussion being clearly distinguished.

Given that certain involved inferences are closely related to the actual observations, we have kept those beside the paragraphs reporting the latter. However, in response to this suggestion we now have a separate Results section that includes all the primary findings and a Discussion section (page 22) that only includes the more general context and implications of the findings in the former section along with possible evolutionary hypotheses.

Reviewer #2 (Evidence, reproducibility and clarity (Required)):The paper by Kaur et al. describes use of bioinformatics to look for relationships between genes encoding cell death proteins in bacteria and metazoans. While the focus is on adaptor proteins related to TRADD-N domains, many other domains are also covered.With the caveat that I am not a bioinformatician or computational biologist, I found this paper to be outstanding. The clear and informative figures provide a rapid way for interested readers to see the composition, origin, and relationships of their favorite proteins.The discussion is intelligent and revealing of evolutionary relationships and plausible explanations of how cell suicide could have been adaptive. They convincingly argue that prokaryotes use these proteins for defense: to detect foreign molecules, transmit signals, and implement responses.The abundance, complexity, and ubiquity of these proteins amongst single-celled organisms suggests defense against other, invading, organisms was an early and very important adaptation during evolution, and perhaps more surprisingly, use of these proteins for the suicide of cells, in order to protect their relatives, was also an early and widely adopted strategy.(Significance (Required)):I am not qualified to verify the bioinformatic analysis.I could not come up with any suggestions for how this manuscript could be improved.

We thank the reviewer for the positive comments on the impact, significance and import of the study.